# Platelet-derived microparticles enhance megakaryocyte differentiation and platelet generation via miR-1915-3p

Mingyi Qu [1,2,3], Xiaojing Zou[1], Fang Fang[1,2,3], Shouye Wang[1,3], Lei Xu[1,3], Quan Zeng[1,3], Zeng Fan[1,3], Lin Chen[1,3], Wen Yue[1,3], Xiaoyan Xie [1,3✉] & Xuetao Pei[1,3✉]

Thrombosis leads to platelet activation and subsequent degradation; therefore, replenishment of platelets from hematopoietic stem/progenitor cells (HSPCs) is needed to maintain the physiological level of circulating platelets. Platelet-derived microparticles (PMPs) are protein- and RNA-containing vesicles released from activated platelets. We hypothesized that factors carried by PMPs might influence the production of platelets from HSPCs, in a positive feedback fashion. Here we show that, during mouse acute liver injury, the density of megakaryocyte in the bone marrow increases following an increase in circulating PMPs, but without thrombopoietin (TPO) upregulation. In vitro, PMPs are internalized by HSPCs and drive them toward a megakaryocytic fate. Mechanistically, miR-1915-3p, a miRNA highly enriched in PMPs, is transported to target cells and suppresses the expression levels of Rho GTPase family member B, thereby inducing megakaryopoiesis. In addition, direct injection of PMPs into irradiated mice increases the number of megakaryocytes and platelets without affecting TPO levels. In conclusion, our data reveal that PMPs have a role in promoting megakaryocytic differentiation and platelet production.

[1] Stem Cells and Regenerative Medicine Lab, Institute of Health Service and Transfusion Medicine Beijing, Beijing 100850, China. [2] Beijing Institute of Radiation Medicine, Beijing 100850, China. [3] South China Research Center for Stem Cell & Regenerative Medicine, SCIB, Guangzhou 510005, China. ✉email: xiex13@126.com; peixt@bmi.ac.cn

Microparticles (MPs) are 100–1000 nm cell-derived membrane-bound small particles. MPs can be secreted by all nucleated cells, as well as erythrocytes and platelets. It has been suggested that MPs act as an important means of message transfer between cells by delivering their containing proteins, DNAs and RNAs[1]. More recently, it has been reported that MPs from macrophages induced macrophage differentiation from naïve monocytes[2] and MPs from megakaryocytes (MK-MPs)[3] promoted (hematopoietic stem/progenitor cell) HSPC differentiation to mature megakaryocytes (MKs). Therefore, we hypothesize that MPs from platelets, such as mature blood cells, play a positive role in piloting stem cells towards platelet parental cells and even platelets. PMPs, unlike MK-MPs, are produced by activated platelets, which provide platelet-specific surface markers and make them the dominant circulating MPs under pathological conditions. As the final product of megakaryocytopoiesis, the activation and consumption of platelets during thrombosis results in the release of large quantities of PMPs[4]. However, little is known about the role PMPs play in regulating thrombopoiesis.

Generally, platelet production is regulated by TPO. Hepatocytes are a major source of TPO. In liver diseases, such as cirrhosis and other advanced liver diseases that present deficient hepatic TPO production, pathological platelet destruction, including shorter platelet survival time and low-grade disseminated intravascular coagulation, has also been reported. Increased levels of circulating PMPs, a result of the increased formation and/or decreased clearance of MPs, have also been reported in these diseases[5–8]. Elevated or normal bone marrow (BM) MK density has been reported in some of the aforementioned diseases[9,10], indicating that megakaryogenesis is not simply suppressed by TPO deficiency in some cases. Conversely, increased numbers of circulating PMPs might account for this observation. PMPs can package cytokines, chemokines and RNA, transfer these bioactive effectors to nucleated recipients and act as vectors of information to regulate the function of target cells[11–13]. Interactions among PMPs and endothelial cells, tumor cells, and immunological cells have been shown in numerous publications[14], and the effect of PMPs on the in vivo proliferation, survival and engraftment of hematopoietic stem cells (HSCs) has also been reported[15,16]. Consequently, a role for PMPs in megakaryopoiesis is proposed.

Our hypothesis is that PMPs induce megakaryopoiesis through a positive feedback mechanism. The activated platelets release MPs, some of which can be internalized by HSPCs and drive the cells toward a megakaryocytic fate. As a consequence, more MKs and platelets can be generated to replace the platelets lost by activation.

In this study, we demonstrate that PMPs can promote MK differentiation and platelet production and reveal the underlying mechanism. The characteristics of PMPs and their ability to mediate MK differentiation are explored. miRNA is one of the abundant contents of PMPs. Elaborated miRNA analysis revealed that, a miRNA highly enriched in PMPs, miR-1915-3p, can be transported to target cells and suppress RHOB expression, thereby induces megakaryocytic differentiation. In vivo and in vitro studies of PMPs suggest that a potential feedback loop is involved in the regulation of platelet numbers, complementary to the typical platelet regulation theory in which physiological and pathological platelet generation is regulated by TPO. The important role of PMPs in MK feedback regulation might expand our knowledge of lineage homeostasis.

## Results

**Megakaryocytosis upon PMP elevation in acute liver injury (ALI).** To confirm the role of PMPs in the regulation of

megakaryocytosis, we chose a mouse carbon tetrachloride ($CCl_4$) toxicity model, which shows pathological manifestations similar to those of human ALI diseases[17,18]. Liver centrilobular necroinflammation and regeneration peaked at day 2 and were nearly resolved by day 8 (Supplementary Fig. 1a, b). No significant changes in hemoglobin levels were observed from day 4, indicating that $CCl_4$-mediated ALI did not cause dehydration or hemoconcentration (Supplementary Fig. 1c). However, the count of platelets in the peripheral blood (PB) was increased significantly on day 8 in ALI group and remained higher than that of control (CON) group until day 12 (Fig. 1a, Supplementary Fig. 1d). This suggested that ALI might cause the release of reserved platelets from the spleen. In that case, the number of platelets in the spleen should be reduced. However, in the spleen, the number of platelets was also increased significantly, but the percentage of the $CD41^+$-cells (megakaryocytic cells) did not show significant alterations on day 8 in ALI group (Supplementary Fig. 1e, f). Thus, the contribution of the spleen to PB platelet upregulation could be excluded.

To determine whether there was definite megakaryocytosis in the BM during ALI, the generation of megakaryocytic cells, was assessed in the BM. Immunohistochemistry and flow cytometry (FCM) analysis demonstrated CD41 was more highly expressed in BM nucleated cells from ALI mice on day 8. Furthermore, the $CD41^+$-cells in the two groups also had morphological differences; their cell size was larger and their nuclei were more lobulated in ALI mice. The percentage of mature megakaryocytes (m-MK, $CD41^+/CD42^+$), late-stage MKs, increased correspondingly. In addition, the percentage of megakaryocyte-erythroid progenitors (MEPs, $CD41^+/CD71^+$), early stage MKs, was higher in ALI mice starting from day 4, and ALI mice also showed more multiploidy cells by day 8 (Fig. 1c, d, Supplementary Fig. 1g, h).

Given the increase in platelets and their progenitor cells in the ALI model, we next addressed the factors that likely contributed to MK differentiation and platelet production. We assessed Tpo mRNA expression in hepatic cells and Tpo protein concentrations in the PB. Due to the increasing necrosis at the prime Tpo synthesis site, Tpo mRNA showed a statistically significant decrease at day 8. Consistently the concentration of PB Tpo was lower in ALI mice throughout the entire process and even reduced to ~70% compared to that of the CON mice on day 8 (Fig. 1e, f), suggesting that Tpo was not the key stimulating factor for platelets in the ALI mice.

Thereafter, we investigated whether the percentage of activated platelets and the number of MPs in the PB of ALI mice increased before the increase in megakaryocytic cell proportion in the BM and the enhancement of platelet count in the PB. As expected, on day 2 after $CCl_4$ administration, we observed that the percentage of $CD62p^+$-platelets and the number of $CD41^+$-MPs increased (Fig. 1g, h). Among the $CD41^+$-MPs, the percentage of $CD62p^+$-MPs, MPs derived from activated platelets, markedly increased (43.9% versus 55.8%) (Fig. 1i). In addition, the concentration of PMPs in the PB was elevated threefold on day 2 (Fig. 1j). Therefore, we are the first to report an increased BM-MKs density following an increased level of circulating PMPs. These discoveries indicate that PMPs may promote MK differentiation.

**PMP internalization promoted megakaryocytopoiesis.** To confirm the function of PMPs, we isolated human and mouse MPs from thrombin-activated platelets and identified them based on the PMPs reported by other researchers in terms of morphology, protein expression and biological functions[19,20] (Supplementary Fig. 2). Then isolated PMPs were stained with PKH67 or PKH26 dye and cocultured with $CD34^+$-HSPCs or leukemic

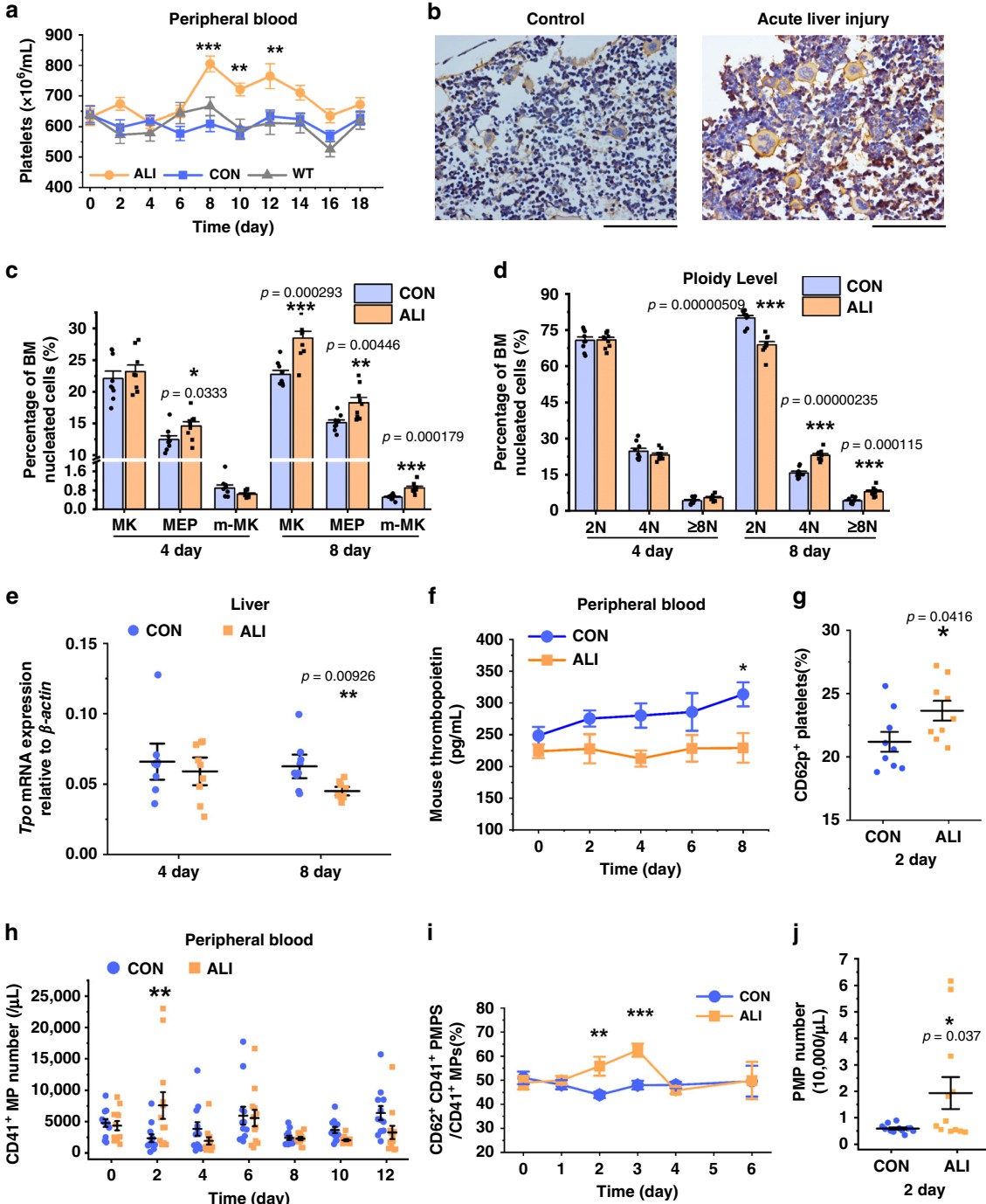

**Fig. 1 Thrombocytosis during ALI was positively correlated with the elevation of PMPs. a** During $CCl_4$-induced acute liver injury (ALI), thrombocytosis was detected as an increased peripheral blood (PB) platelet counts in the ALI vs. control (CON) mice ($n = 12$ mice, two-way repeated-measures ANOVA with Bonferroni's multiple comparison tests, $p = 0.0045$ (Con as control)). Non-treated mice (WT) were set as baseline control. **b**, **c** Megakaryocytopoiesis in bone marrow (BM) after $CCl_4$ injection is shown. On day 8 after injection, representative femur sections stained with CD41 and hematoxylin showed increased levels of CD41-positive cells (Scale bars: 100 μm) (**b**), and a histogram indicated that BM whole megakaryocytic cells (MK, CD41 positive), including mature megakaryocytes (m-MK, CD41, and CD42 double-positive) and megakaryocyte-erythroid progenitors (MEP, CD41, and CD71 double positive), were increased in the ALI mice **c**. Representative FCM plots are shown in Supplementary Fig. 1g. **d** The percentage of cells of each ploidy in BM is shown. **e**, **f** Liver TPO synthesis was suppressed under ALI, as demonstrated by liver cell qPCR (**e**) and plasma TPO ELISA analyses (two-way repeated measures ANOVA with Bonferroni's multiple comparison tests, p = 0.0085) (**f**). **g** PB-PLTs incorporated increased levels of CD62p-positive activated PLT in the ALI mice on day 2 after injection. **h–j** Platelet-derived microparticles (PMPs) increased in the ALI mice ($n = 12$ mice). Compared with those of the control group, the circulating CD41+ microparticles (MPs) increased (two-way repeated-measures ANOVA with Bonferroni's multiple comparison tests) (**h**), and the proportion of CD41+/CD62p+ PMPs among the CD41+ MPs in PB increased as well (two-way repeated-measures ANOVA with Bonferroni's multiple comparison tests, $p = 0.0109$) (**i**). The absolute count of PMPs on day 2 after $CCl_4$ injection was higher than that in the control group when platelets started to increase mice) (**j**). Each time point in each animal cohort represents a mean ± S.E.M. of nine mice and two-tailed unpaired $t$-tests were used unless otherwise specified. Source data are provided as a Source Data file *$P < 0.05$, **$P < 0.01$, ***$P < 0.001$.

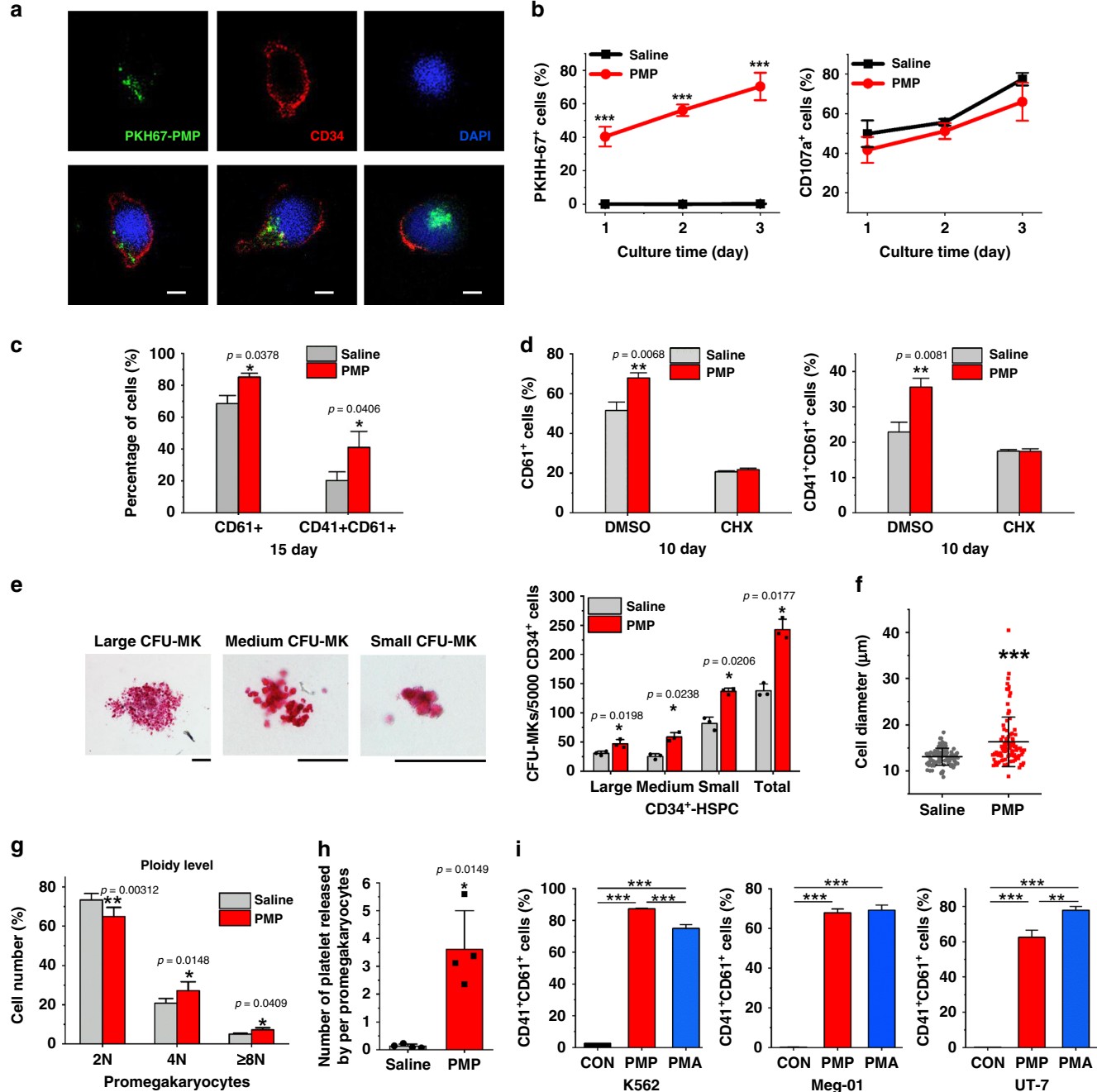

cells to check the internalization of PMPs. To exclude false-positive signals arising from cells engulfing ultracentrifugation-induced PKH26 nanoparticles, aggregated PKH26 fluorescent dye collected directly through the same centrifugation method used for PMPs was used as another control[21]. We observed that more than 60% of leukemic cells had PKH26 fluorescence of varying intensity after 24 h of PMP application (Supplementary Fig. 3a, b). It took 72 h for nearly 80% of CD34+-HSPCs to internalize the PMPs, probably because the HSPCs have a characteristically small size and slow proliferation. PMPs were clustered in CD34+-HSPCs and megakaryocytic cells afterward. The marker of PMPs, CD107a, showed no significant difference between groups with and without PMP treatment. Recipient cells did not express the same surface markers as PMPs but emitted fluorescence like PMPs after labeling, indicating that PMPs did not attach to the surface of K562 cells or HSPCs but penetrated into the cells.

Accordingly, it was speculated that the surface markers could not be passed to the target cells directly by the entry and internalization of PMPs. (Fig. 2a, b, Supplementary Fig. 3d).

After we demonstrated that the isolated PMPs were internalized by the cells, their effect on HSPC-MK differentiation in vitro was evaluated. PMP supplementation at the initial time point resulted in elevated expression of the MK biomarkers until the terminal phase of maturation (Fig. 2c, Supplementary Fig. 4a). The same effect was also observed in heterogeneous cord blood (CB) MNCs, which are a mixture of HSCs and progenitor cells (Supplementary Fig. 4b). To assess the actual expression of CD41 by MNCs vs. the association of PMPs with MNCs, a protein synthesis inhibitor cycloheximide, was used during megakaryocytic induction. In the presence of cycloheximide, PMP supplementation did not result in a significant alteration in the percentages of the CD61+ and CD41+ populations (Fig. 2d). In

**Fig. 2 Addition of PMPs promoted megakaryocyte differentiation and polyploidization. a** Confocal microscopy imaging demonstrated the internalization of isolated PMPs (green) into CD34+-HSPCs (cytoplasm, red; nuclei, blue) (top: monochrome, bottom: merge. Incubation time: 72 h. Scale bars: 5 μm). **b** FCM analysis of CD34+-HSPCs showed that the majority of the cells internalized the supplemented PMPs stained with PKH67. PMPs did not apparently alter CD34+-HSPC surface marker (CD107a) expression during a 72 h culture in MK induction medium ($n = 3$ donors, two-way repeated-measures ANOVA with Bonferroni's multiple comparison tests, $p < 0.0001$). **c** PMPs promoted the proportion of CD61+ and CD41+/CD61+ cells on day 15. Representative FCM plots are shown in Supplementary Fig. 4a (mean ± S.E.M. of $n = 7$ donors). **d** Cycloheximide (CHX) inhibited the PMP-mediated promotion of the proportion of CD61+ and CD41+/CD61+ cells on day 10. (mean ± S.E.M of three donors, two-way repeated-measures ANOVA with Bonferroni's multiple comparison tests). **e** PMPs expanded CFU-MK colonies from 72 h-treated CD34+-HSPCs. Colonies were stained with human CD41 antibody, and the size was calculated (Scale bars: 100 μm). ($n = 3$ donors) **f** Twenty days after PMP internalization, megakaryocytes derived from the CD34+-HSPCs showed enlarged cell dimensions. Cytospin cells were stained with Wright-Giemsa solutions and measured from five random views. The error bars represent the standard deviation. (Two-tailed unpaired $t$-tests with Welch's correction, $p < 0.0001$). **g** The percentage of cultured CD61+-promegakaryocytes five days after PMP internalization in each ploidy is shown. Representative FCM plots are shown in Supplementary Fig. 4e. ($n = 4$ donors) **h** The number of platelets released in the culture media normalized to each CD61+-promegakaryocyte is shown. Representative FCM plots are shown in Supplementary Fig. 4f. ($n = 4$ donors) **i** FCM analysis of the megakaryocytic lines K562, Meg-01 and UT-7 showed that PMP supplementation demonstrated similar or even better megakaryocytic promotion effects than PMA treatment. Megakaryocyte differentiation was determined by the CD41+/CD61+ proportions. Representative FCM plots are shown in Supplementary Fig. 3c ($n = 3$ independent experiments, one-way ANOVA with Bonferroni's multiple comparison tests, $p < 0.0001$). All data are expressed as mean ± S.D from two-tailed paired-samples $t$-tests, $P$-values unless otherwise specified: Source data are provided as a Source Data file *$P < 0.05$, **$P < 0.01$, ***$P < 0.001$.

addition to surface marker expression, a more than 1.75-fold increase in the number of total MK colony-forming units (CFU-MKs, 242.67 versus 138) with PMP internalization, among which all the numbers of small, medium and large CFU-MKs increased significantly (Fig. 2e), suggesting a positive effect of PMPs at each stage of megakaryocytic differentiation, since the size of CFU-MK colonies represents the megakaryocytic potential of progenitor cells. The PMP-internalized MNCs also had more CFU-MKs of all sizes (Supplementary Fig. 4c). When comparing MK maturity by cellular morphology, CD34+-HSPCs treated with PMPs showed more mature MK phenotypes and a larger diameter (Fig. 2f), and CD61+-pro-MKs treated with PMPs showed more polyploid cells (4 N and 8 N or more) (Fig. 2g, Supplementary Fig. 4e). Concomitantly, we evaluated the number of platelets released into the culture medium. From the same number of seeded pro-MKs, we obtained more platelets in pro-MKs treated with PMPs than the control (3.61 verse 0.13 platelets per pro-MK) (Fig. 2h, Supplementary Fig. 4f).

Three megakaryocytic cell lines, K562, UT-7 and Meg-01, which are commonly induced into cells with megakaryoblastic characteristics with phorbol myristate acetate (PMA), were also chosen to confirm PMP function. After PMPs supplementation, more than 60% of the cells became CD41+/CD61+. In Meg-01 cells, no significant difference between PMP and PMA treatment was observed, and in K562 cells, the induction rate under PMP treatment was even higher than that observed for the PMA-positive control (Fig. 2i, Supplementary Fig. 3c).

Our study demonstrated the important role that PMPs play in MK differentiation, endomitosis and platelet production, which is similar to TPO. Therefore, we checked the TPO protein concentration in isolated PMPs. Only a tiny amount of TPO was packaged into the membrane structures (~0.5 pg per $1 \times 10^5$ PMPs) compared to the content in MK induction medium (100 ng per mL) (Supplementary Fig. 5a). To determine the role of TPO inside PMPs, we removed exogenously added TPO and treated the cells with excess antibodies against TPO or its receptor. Under such circumstances, the promoting effect of PMPs on the percentage of CD61+ and CD41+/CD61+ populations derived from MNCs remained significantly higher (Supplementary Fig. 5b), indicating that TPO packaging in the PMPs did not fully explain their effect on MK differentiation.

**miR-1915-3p is one of the key miRNAs transferred by PMPs.** PMP-encapsulated bioactive substances play a role in promoting MK differentiation and maturation. Since TPO was excluded

from the functional contents of PMPs, RNA components were explored. Based on previous reports, MPs are major transport vehicles for distinct miRNAs in the circulation, and miRNAs in MPs play an important role in intercellular communication[2,22]. Accordingly, we narrowed our analysis of potential functional components in PMPs to miRNAs and identified the miRNA content of PMPs.

Collectively, 182 miRNAs were detected in the isolated human PMPs from five independent donors, although individual differences in miRNA expression were also observed. Thereafter, these miRNAs were ranked by expression level, and those expressed above the median level from each unique subject were considered to be abundantly expressed miRNAs. The intersection of abundantly expressed miRNAs in PMPs from each donor yielded 48 commonly expressed miRNAs in PMPs, as shown in Venn diagram and heat map (Fig. 3a, b). Among these 48 miRNAs, miR-4454 ranked number 1, and its level was consistent in all samples examined. Then, the other 24 miRNAs that were expressed at levels higher than or equal to the median level of the remaining 47 miRNAs were subjected to further examination.

As previously reported, PMPs could transfer miRNA molecules to recipient cells, and the enhancement of definitive miRNAs was investigated in MNCs that received PMPs. Although the miRNA microarray showed that miR-4454 was expressed at a high and stable level in all human PMP samples, PMP internalization did not result in a significant alteration of miR-4454 in MNCs. Therefore, for PMP-treated cells, miR-4454 and cell number were used to normalize miRNA expression. We identified 6 miRNAs among the top 25 miRNAs enriched in human PMPs that were increased in CB-MNCs after PMP internalization. Overall, 6 miRNAs (miR-1915-3p, miR-223, let-7a-5p, miR-15b-5p, miR-3960, and miR-2861) were highly enriched in human PMPs and were capable of being transported to target cells (Fig. 3c).

The 6 screened miRNAs together with the expression control miR-4454 were transiently overexpressed in K562, UT-7, and Meg-01 cells to determine whether those miRNAs played a role in controlling MK differentiation. A similar trend for the same miRNAs in different cell lines was discovered. Among the 7 miRNAs, miR-1915-3p overexpression showed the most profound and statistically significant promoting effect on the expression of MK surface markers. miR-223 was the second functional miRNA that increased the expression of megakaryocytic surface markers (Fig. 3d).

To determine whether the miRNA content of PMPs varied in the mouse model of thrombocytosis, we detected the expression of the

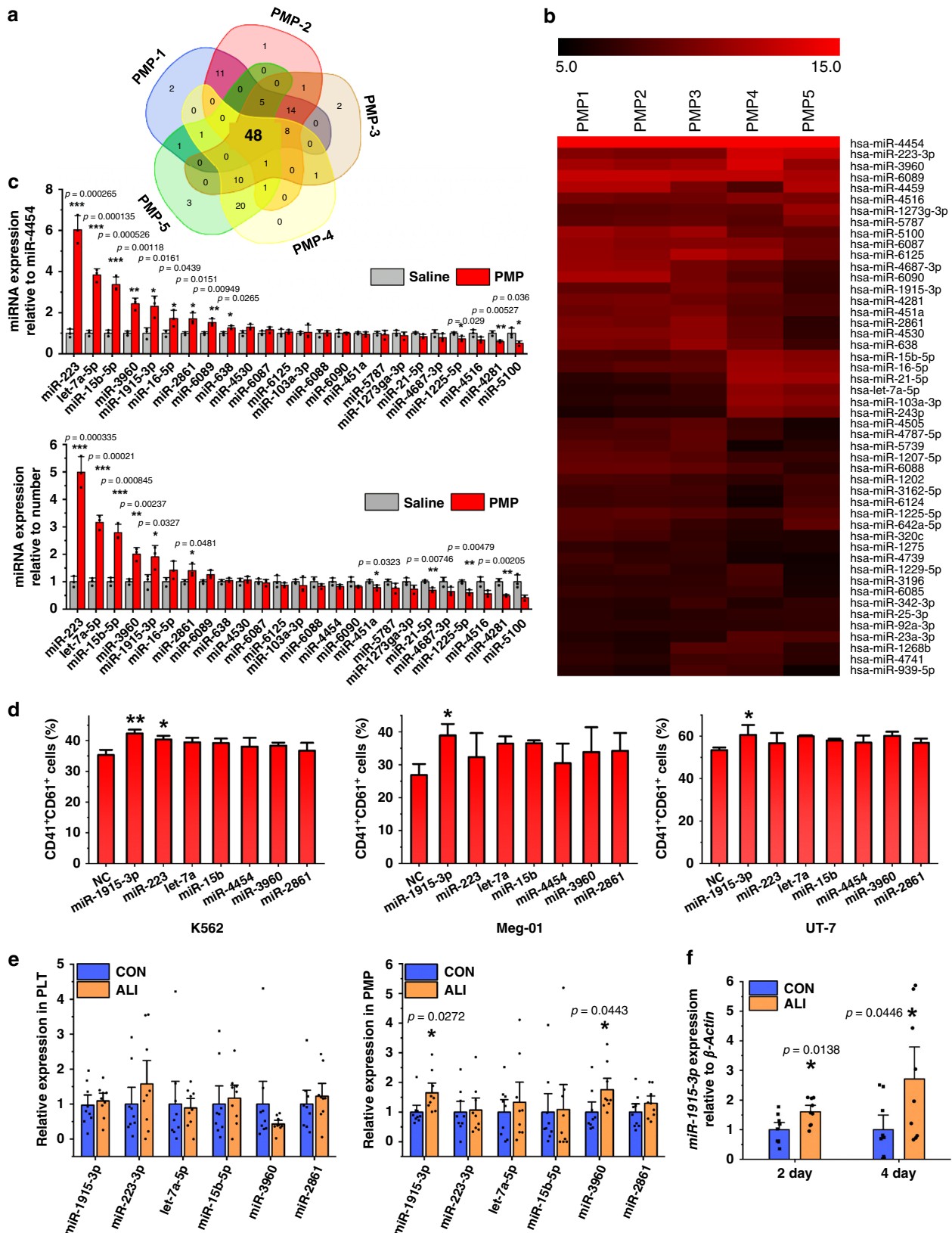

above-proven highly enriched miRNAs in mouse platelets and PMPs. miR-4454 was discovered only in humans, and the 6 selected miRNAs were analyzed. The level of miRNAs in ALI mouse platelets was consistent with that in normal mice. In contrast, miR-1915-3p and miR-3960 were upregulated in PMPs from ALI mice (Fig. 3e). The clear augmentation of PMPs together with the

upregulation of miR-1915-3p content in PMPs might increase the amount of transferable miR-1915-3p during ALI. Thus, we compared the expression of miR-1915-3p in the BM nucleated cells of ALI mice versus CON mice and found an upregulation of miR-1915-3p by 1.6-fold (2 days) and 2.7-fold (4 days) after $CCl_4$ treatment (Fig. 3f). The observation in vivo validated the coherence

**Fig. 3 Profile and verification of PMP-containing miRNAs. a** Venn diagrams were generated for the highly expressed miRNAs in all five subjects. **b** Heatmaps of the highly expressed miRNA contents of the PMPs from five different donors are shown. **c** Relative expression of the 25 most abundant miRNAs in the PMPs was analyzed by qPCR. miRNA expression is shown relative to miR-4454 (top panel) and the cell number (bottom panel) ($n = 3$ donors) **d** The cells were transfected with 7 miRNA mimics sorted from the expression profile and cultured in medium containing 1 nM PMA to induce megakaryocytic differentiation. The impact of miRNA overexpression on these megakaryocytic cell lines was determined by FCM. Altered expression of the miRNAs influenced the expression of the MK integrins CD41 and CD61 ($n = 3$ independent experiments, one-way ANOVA with Dunnett's test, NC as control). **e** Relative expression of the 6 selected miRNAs in mouse platelets (left) and PMPs (right) 2 days after $CCl_4$ injection was analyzed by qPCR. miRNA expression is shown relative to the platelet or PMP number ($n = 9$ mice). **f** qPCR analysis of miR-1915-3p expression in BM-nucleated cells of the ALI mouse ($n = 9$ mice). All data are shown as the mean ± S.D from two-tailed unpaired-samples $t$-tests unless otherwise specified. $P$-values. Source data are provided as a Source Data file *$P < 0.05$, **$P < 0.01$, ***$P < 0.001$.

between increased PMPs and upregulated miR-1915-3p in target cells, which was discovered in vitro with human cells.

**miR-1915-3p shows positive effect on MK differentiation**. The effect of miR-1915-3p on hematopoiesis has not been previously reported. To confirm the role of miR-1915-3p in MK differentiation, we first investigated the endogenous expression of miR-1915-3p in HSPC-MK differentiation. The expression of miR-1915-3p was low in HSPCs and progressively increased with MK differentiation (Supplementary Fig. 6a), suggesting that miR-1915-3p might be involved in MK differentiation. Thereafter, we overexpressed miR-1915-3p in HSPCs to determine whether the upregulation of miR-1915-3p play a role in controlling MK differentiation. Transfection of miR-1915-3p mimics increased miR-1915-3p levels ~200-fold (Fig. 4a). Subsequently, we evaluated the effect of miR-1915-3p overexpression on MK colony formation and MK marker expression. For CD34+-HSPCs, miR-1915-3p overexpression resulted in a statistically significant increase in total CFU-MKs, especially in medium and small CFU-MKs, but showed no effect on large CFU-MKs, suggesting that there may be other components of PMPs that promote MK differentiation at a very early stage of differentiation (Fig. 4b). Moreover, miR-1915-transfected CD34+-HSPCs derived much larger CD61+ and CD41+/CD61+ populations (Fig. 4c, Supplementary Fig. 6c). The cell size and ploidy level were increased together with the overexpression of miR-1915-3p (Fig. 4d). A similar effect was also discovered in miR-1915-3p-overexpressing CB-MNCs (Supplementary Fig. 6d–f).

Subsequently, stable miR-1915-3p overexpression or downregulation in megakaryocytic cell lines was established and used to confirm the positive role of miR-1915-3p in MK differentiation. Successful overexpression of miR-1915-3p is shown in Fig. 4e. Cells with exogenous miR-1915-3p were then cultured in medium containing PMA. Most of the induced cells resulting from transfection and PMA treatment were CD61-positive, overexpression of miR-1915-3p significantly promoted the percentage of CD41+/CD61+ cells in all megakaryocytic cell lines (Fig. 4f, Supplementary Fig. 7a). Another indicator of MK maturation is polyploidization. The proportion of polyploid cells (≥8 N) in pc3.0-pri-miR-1915-3p cells was much greater than that in pc3.0 (blank vector) cells (Fig. 4g, Supplementary Fig. 7b, c). By transfection with pcDNA3.0-pri-miR-1915-3p-sponge-neomycin (sponge), miR-1915-3p expression was successfully reduced to ~40% in all megakaryocytic cell lines (Fig. 4h). As expected, the miR-1915-3p sponge retarded MK differentiation. The percentages of CD41+/CD61+ populations in K562, UT-7 and Meg-01 cells stably transfected with miR-1915-3p sponges were 34.9%, 42.3% and 35.7%, whereas in the control cells (pcDNA 3.0), the percentages were 48.5%, 73.3% and 52.8%, respectively (Fig. 4i, Supplementary Fig. 7d).

**RHOB is a target of miR-1915-3p and PMP function mediator**. The target miR-1915-3p has not been previously reported.

However, in silico prediction of miRNA targets using three databases (TargetScan, microRNA.org, and PITA) suggested that 1359 mRNAs could be regulated by miR-1915-3p at the transcriptional or posttranscriptional level. We reasoned that the target genes of miR-1915-3p in PMPs carried a miR-1915-3p complementary site in their 3′UTR, as predicted, and were downregulated after PMP addition. To identify the genes inversely regulated by PMPs, we performed a microarray for global gene expression profiling in CB-MNCs with/without PMP treatment. In MNCs treated with PMPs, 129 genes were downregulated. Cataloged by enrichment of Gene Ontology biological processes, these genes were found to be highly enriched in pathways related to blood coagulation, innate immune response, immune response, inflammatory response, and platelet activation, all of which play important roles in the function of platelets (Table 1)[23]. This confirmed again that PMPs played a role in controlling MK development. Subsequently, the intersection between bioinformatics-predicted miR-1915-3p targets and microarray-detected downregulated mRNAs identified 9 genes that might mediate the function of miR-1915-3p in MK development (Fig. 5a). To validate the microarray finding, quantitative real-time polymerase chain reaction (qPCR) for the 9 genes was conducted with MNCs from another five donors and leukemia cells with or without PMPs. As shown in Fig. 5b, only part of the MNC sample's qPCR was in good agreement with the microarray data, but upregulation of some selected *genes* was also observed in the other MNCs and K562, Meg-01 and UT-7 cells after PMP treatment. *RHOB* was the only gene that was consistently downregulated in all cells that took up PMPs. Thus, we ultimately narrowed our analysis of potential target genes of PMPs and miR-1915-3p to *RHOB*, which might be involved in the common process of MK differentiation. Similarly, protein expression of RHOB coincidently decreased following PMP treatment (Fig. 5c). Furthermore, a negative correlation between PMP treatment and *RHOB* expression was found.

We next sought to confirm the direct regulation of *RHOB* by miR-1915-3p. By performing integrative bioinformatic analysis, we discovered a complete complementary site to the miR-1915-3p seed sequence in the 3′-UTR of *RHOB*. Using luciferase reporter assay, we confirmed that miR-1915-3p targeted the putative target sites from *RHOB* 3′-UTR (Fig. 5d). *RHOB* expression was downregulated at the mRNA and protein levels in cells subjected to miR-1915-3p overexpression (Fig. 5e) and upregulated in cells subjected to miR-1915-3p knockdown (Fig. 5f). The in vivo study also demonstrated that in ALI mice, which showed increased circulating PMP levels and BM miR-1915-3p expression, the protein and mRNA levels of *Rhob* in BM nucleated cells were considerably lower than those in CON mice (Fig. 5g). Therefore, we confirmed that *RHOB* expression inversely correlated with miR-1915-3p expression. A few previous reports[24,25] and our study[26] demonstrated that *SCR* and *ROCK*, downstream effectors of *RHOB*, acted as negative regulators of MK differentiation. Accordingly, we deduced that miR-1915-3p carried by PMPs was transported to target cells and suppressed

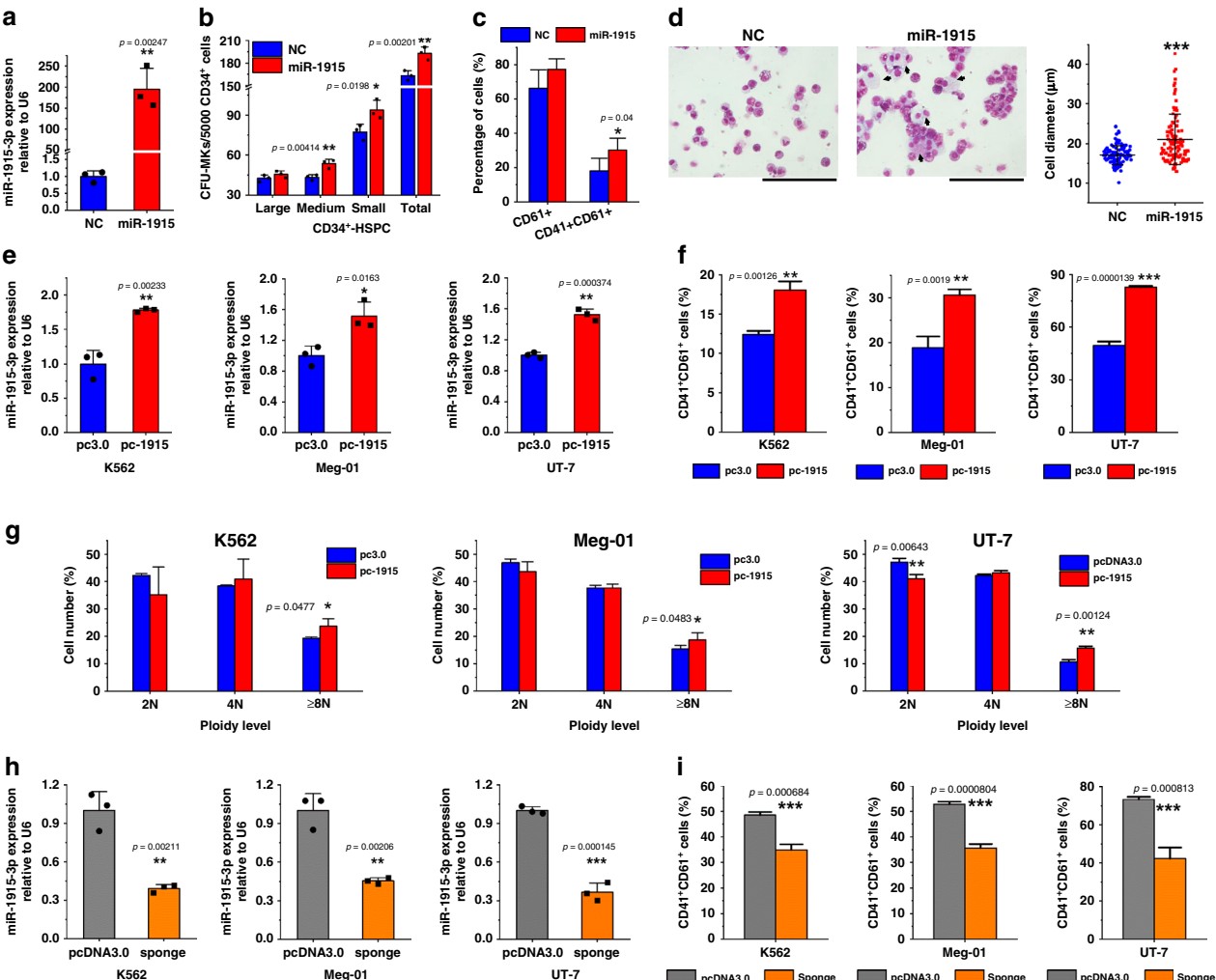

**Fig. 4 Alteration of the miR-1915-3p expression affected MK differentiation and maturation. a** qPCR confirmed the overexpression of miR-1915-3p in CD34[+]-HSPCs after miRNA mimics (miR-1915) were transfected for 24 h compared with the negative control (NC). U6 was set as the normalized control ($n = 3$ donors). **b, c** Overexpression of miR-1915-3p promoted the CFU-MKs from the CD34[+]-HSPCs ($n = 3$ donors, paired-samples $t$-tests) **b** and increased the expression of the MK integrins CD41 and CD61 on day 15. (mean ± S.E.M. of $n = 5$ donors, paired-samples $t$-tests) (**c**). Representative FCM plots are shown in Supplementary Fig. 6c. **d** By megakaryocytic induction on day 20, Cytospin and Wright–Giemsa staining demonstrated more polyploid cells and larger cell dimensions (black arrows) with miR-1915-3p overexpression. Cells from five random views were measured. Statistical analysis and results were conducted on data from three biologically independent experimental replicates Unpaired $t$-tests with Welch's correction were used for the statistical analysis. $p < 0.0001$. (Scale bars: 100 μm) **e–g** In the K562, Meg-01 and UT-7 cells, overexpression of miR-1915-3p by stable transfection with a plasmid vector carrying primary miR-1915-3p ($n = 3$ independent experiments) (**e**) promoted the CD41[+]/CD61[+] populations ($n = 3$ independent experiments) (**f**) when cells were treated with a suboptimal amount of PMA for 3 days. Representative FCM plots are shown in Supplementary Fig. 7a. Cell polyploidization was upregulated with overexpression as well. **g** The percentage of cells of each ploidy is shown. Representative FCM histograms are shown in Supplementary Fig. 7b ($n = 3$ independent experiments). **h, i** Suppression of miR-1915-3p expression with a miRNA sponge ($n = 3$ independent experiments) **h** significantly decreased the CD41[+]/CD61[+] populations in the PMA-treated K562, UT-7 and Meg-01 cells **i** Representative FCM plots are shown in Supplementary Fig. 7d ($n = 3$ independent experiments). All data are shown as the mean ± S.D from two-tailed unpaired-samples $t$-tests. P-value unless otherwise specified. Source data are provided as a Source Data file *$P < 0.05$, **$P < 0.01$, ***$P < 0.001$.

*RHOB* expression in target cells, thereby inducing megakaryocytic differentiation and maturation (Supplementary Fig. 8).

**PMP transfusion stimulates megakaryocytopoiesis in vivo.** Mice irradiated with a semilethal dose were chosen to validate the function of PMPs in vivo. The hematopoietic system is compromised at this dose, leading to fewer HSPCs. However, the surviving HSPCs are capable of hematopoietic reconstitution. We isolated PMPs from green fluorescent protein (GFP) transgenic mice, verified that more than 70% of PMPs expressed GFP and transfused these GFP-PMPs into mice immediately after

irradiation to trace the homing and uptake of PMPs (Supplementary Fig. 9a, b). The expression of GFP was markedly elevated in Sca-1 (stem-cell antigen-1)-positive BM nucleated cells on day 4 after GFP-PMP injection, which indicates that PMPs entered the BM and were internalized by Sca-1-positive nucleated cells (Fig. 6a).

We next checked the effect of PMPs on megakaryocytic differentiation and platelet production. A colony-forming assay was used to assess the effects of PMP internalization on the differentiation of nucleated cells derived from the BM of irradiated mice. Consistent with the in vitro observations, we found more CFU-MKs (36.25 versus 19.25) from mice injected

**Table 1 Gene expression profiling of MNCs in the presence or absence of PMP treatment.**

| Molecular and cellular function | Fold enrichment | P-value | Number of molecules | FDR |
|---|---|---|---|---|
| Blood coagulation | 1.739779 | 1.24E−09 | 184/486 | 4.80E−06 |
| Innate immune response | 1.538794 | 1.30E−08 | 257/770 | 2.51E−05 |
| Immune response | 1.834004 | 2.47E−08 | 130/324 | 3.18E−05 |
| Inflammatory response | 1.811905 | 3.40E−08 | 132/333 | 3.28E−05 |
| Platelet activation | 1.966704 | 2.15E−07 | 92/213 | 0.000166 |
| Signal transduction | 1.419512 | 4.08E−07 | 300/974 | 0.000263 |
| Cytokine-mediated signaling pathway | 1.797146 | 1.24E−06 | 103/261 | 0.000685 |
| Type I interferon signaling pathway | 2.691139 | 1.74E−06 | 41/69 | 0.000843 |
| Adaptive immune response | 2.064937 | 6.37E−06 | 61/134 | 0.002737 |
| Response to lipopolysaccharide | 1.979213 | 9.54E−06 | 65/149 | 0.003686 |

The functional summary of the differentially expressed mRNAs (>1.2-fold or <0.833-fold changes) is shown. The PANTHER Classification System (http://pantherdb.org/) was used for Gene Ontology analysis.

with exogenous PMPs (Fig. 6b). Significantly higher proportions of CD41$^+$ cells from irradiation day 4 to day 21 and more polyploid megakaryocytic cells on day 21 in the BM reflected that MK differentiation in vivo was promoted by PMPs (Fig. 6c, d, Supplementary Fig. 9f). MKs were detectable in the PB because of changes in vascular permeability after irradiation. Higher proportions of whole megakaryocytic cells and m-MK were also detected in the PB of mice injected with PMPs throughout the 21 days, suggesting that PMPs promoted MK differentiation and maturation (Fig. 6e, Supplementary Fig. 9g). Time course studies revealed reductions in blood cells and platelets after irradiation at a semilethal dose, and the platelet decline was ~90% on day 8. As expected, PMPs enhanced the recovery of platelets when transfused into mice. Around days 10–16, platelet counts were elevated by 50% in mice injected with PMPs compared to those in mice injected with normal saline, but the count of white blood cell and red blood cell did not increase (Fig. 6f, Supplementary Fig. 9h). Though the plasma Tpo concentration was increased after irradiation, it remained unaffected in PMP-treated mice compared with the level in normal saline-injected mice throughout the 3-week observation period (Fig. 6g). Thus, we demonstrated that intravenous injection of PMPs promoted MK differentiation and platelet production in vivo.

Subsequently, we compared the expression of miR-1915-3p and *Rhob* in BM nucleated cells from PMP-treated mice. Compared to the Saline treatment, 4 days after exogenous PMP injection, miR-1915-3p was upregulated by 1.6-fold, and *Rhob* expression was downregulated by 0.6-fold at both the protein and mRNA levels (Fig. 6h, i).

To directly validate that the packaging and transfer of miR-1915-3p by PMPs positively affect MK differentiation and platelet production, miR-1915-3p-overexpressing PMPs were created and transfused ex vivo. In brief, mouse platelets were transfected with fluorescent miRNA mimics (FAM-miR-1915 or FAM-NC). Fluorescent PMPs were isolated from these FAM platelets and transfused into irradiated mice (Supplementary Fig. 10a–c). Subsequently, PMPs with enhanced miR-1915-3p were injected into irradiated mice. Higher proportions of whole megakaryocytic cells and MEPs in the BM and an increased count of platelet in the PB were detected on day 14, suggesting that overexpression of miR-1915-3p in PMPs promoted the expression of MK integrins and the production of platelets in vivo (Fig. 6j, k, Supplementary Fig. 10d).

## Discussion

An increasing number of studies have shown that PMPs participate in multiple physiological and pathological functions and have mostly focused on their involvement in cancer development and immune processes and their biological contents for functional mechanisms. Our findings first revealed the distinct role of PMPs in regulating thrombopoiesis and the mechanisms underlying this regulation. PMPs transferred miR-1915-3p to HSPCs to downregulate the expression of *RHOB* and drive the cells toward a megakaryocytic fate to produce platelets. This regulatory system defines an unrecognized feedback pathway between platelet consumption and replenishment.

In this study, a specific miRNA profile of PMP capsules was discovered, and 182 miRNAs were found in PMPs. PMPs have been reported to be major transport vehicles that transfer miRNA to cells in vitro and in vivo and modulate target cell gene expression[19,27]. In agreement with these studies, we confirmed that the miRNA molecules contained in the PMPs were shuttled not only to primary hematopoietic cells but also to other megakaryocytic cell lines. We and others found that many PMP-containing miRNAs were significantly increased in PMP-treated cells[28,29]. One of the defined miRNAs, miR-223, which might enhance MK differentiation and maturation through repression of *MYH10* and *LMO2*[30,31], was significantly upregulated in PMP-treated cells. In contrast, transfection of miR-1915-3p, a highly enriched miRNA in PMPs, showed a more significant effect than miR-223 in promoting MK differentiation. Previous studies indicated that miR-1915-3p might be implicated in the process of oxidative stress, tumorigenesis and disease development, but the functions of miR-1915-3p remain largely unproven[32–35]. In our study, the role of miR-1915-3p was clarified by miRNA gene modification. Based on our findings, the positive effect of miR-1915-3p on megakaryopoiesis was shown. In addition to miR-1915-3p and miR-223, the roles of the rest of the miRNAs in PMPs and their cooperative effects remain to be determined in further studies.

Analysis of the gene expression profile of PMP-treated cells revealed a number of cellular processes, including blood coagulation, immune and inflammatory responses, and platelet activation. We also demonstrated that the addition of PMPs elevated the levels of miR-1915-3p and suppressed the expression of *RHOB* during MK differentiation. The direct regulation of *RHOB* by miR-1915-3p and PMPs was further confirmed by qPCR, Western blotting and luciferase reporter assay. In this way, a direct target of miR-1915-3p was identified and experimentally validated. *RHOB* is a Rho family GTPase implicated in a variety of cellular processes, including cell proliferation, DNA repair and apoptosis[36]. The *Rho/ROCK* pathway is well acknowledged as a negative regulator of MK endomitosis and proplatelet formation[24,37]. *RHOB* activated *SCR* and *ROCK*[24,25], and inhibitors of *SCR* and *ROCK* promoted MK differentiation and maturation[26]. Accordingly, PMPs contained high levels of miR-1915-3p and might induce MK differentiation of recipient cells by inhibiting the Rho/ROCK pathway.

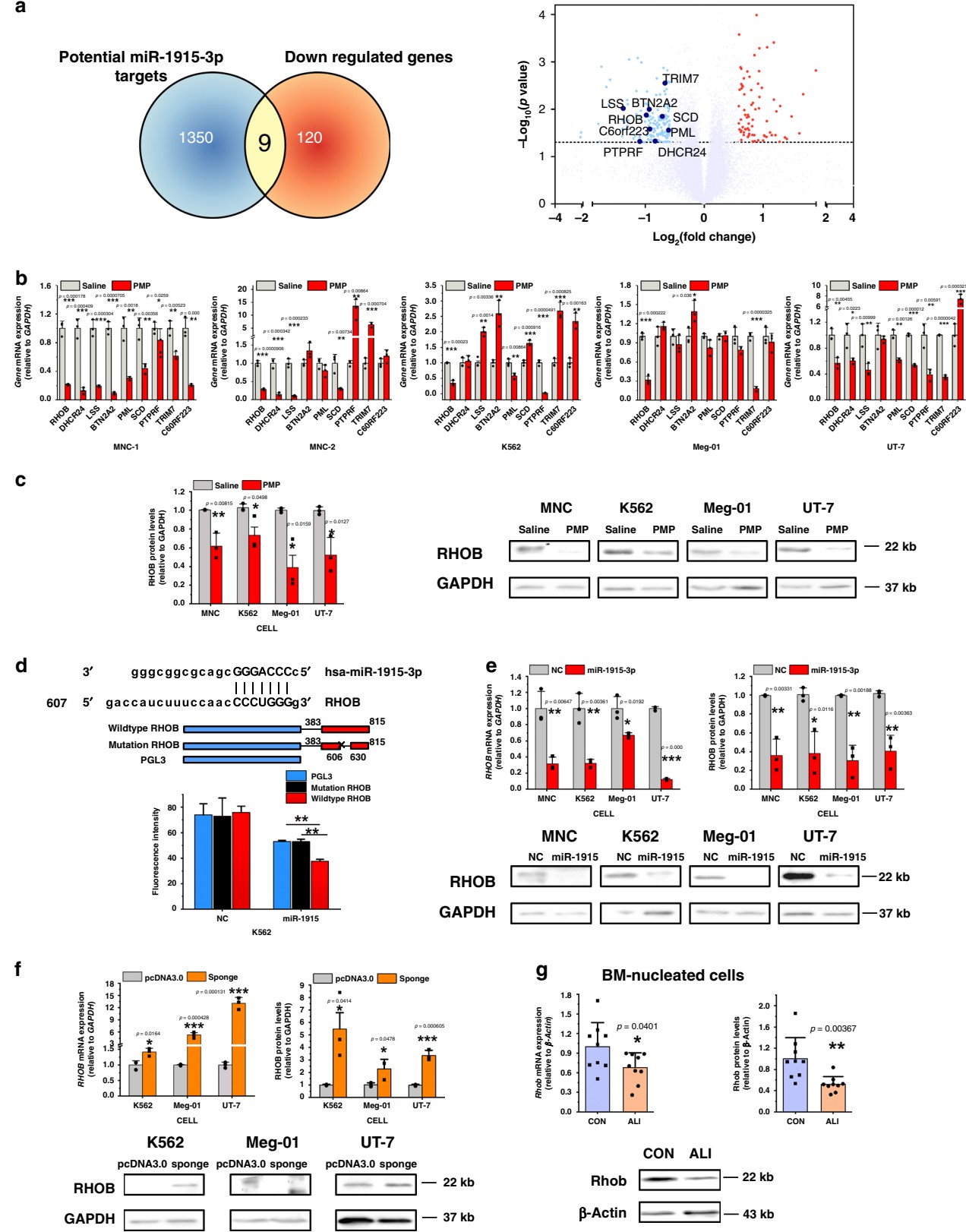

MPs are secreted by all nucleated cell types, including MKs. Flaumenhaft and colleagues reported the existence of MK-MPs in the circulation[38]. MK-MPs are CD41+ like PMPs; therefore, major circulating CD41+ MPs are derived from 2 sources: activated platelets and MKs[38,39]. CD41+ MPs constitute ~70–90% of circulating MPs[40]. However, the ratio of PMPs and MK-MPs in

CD41+ MPs is still controversial. Some studies have suggested that the majority of PB microvesicles were derived from platelets[2,40,41], but a single study from Jinlin Jiang et al. claimed that MK-MPs were the most abundant MPs in the circulation[3,42]. The different physiological states of individuals might alter the components of MPs. Unlike MKs that constitutively produce

**Fig. 5 _RHOB_ is a direct target of miR-1915-3p and the mediator of PMP function. a** Differential expression of the potential target genes of PMPs and miR-1915-3p is shown. The selected 9 _genes_ are shown in the bottom panel, which were downregulated consistently and significantly (<0.667-fold changes and $p < 0.05$) with PMP treatment in gene expression profiling analysis and potentially interacted with miR-1915-3p through bioinformatic analysis (Venn diagram, upper panel). **b** qPCR analysis of potential target _gene_ expression in the MNC, UT-7, K562 and Meg-01 cells. _RHOB_ was consistently downregulated in all cells detected (normalized to Saline, $n = 3$ independent experiments per condition). **c** Quantification of the RHOB protein levels by Western blotting analysis (normalized to Saline, $n = 3$ independent experiments per condition). Lower panel: Representative Western blotting images of RHOB expression. RHOB was suppressed with PMP treatment. **d** Upper, putative miR-1915-3p binding sequence in the _RHOB_ 3′-UTR. Middle, the scheme of the luciferase reporter vector with the wild-type miR-1915-3p recognition fragment from the _RHOB_ 3′-UTR or mutant recognition site. miR-1915-3p overexpression decreased the luciferase activity of the reporter containing wild-type binding sequence of RHOB compared with that of the reporter containing the mutant form and the controls ($n = 3$ independent experiments, one-way ANOVA with Bonferroni's multiple comparison test, $p = 0.0009$). **e** _RHOB_ was downregulated in the K562, Meg-01, UT-7, and MNC cells transiently transfected with miR-1915-3p as evaluated by qPCR (left) and Western blots (right) (normalized to NC, $n = 3$ independent experiments per condition). **f** _RHOB_ was upregulated in the K562, UT-7 and Meg-01 cells stably transfected with a miR-1915-3p sponge as evaluated by qPCR (left) and Western blots (right) (normalized to pcDNA3.0, $n = 3$ independent experiments per condition). **g** _Rhob_ was downregulated in the ALI mouse BM nucleated cells as evaluated by qPCR (left) and Western blotting (right) (normalized to CON mouse, $n = 9$ mice). All data are expressed as the mean ± S.D, P-value. Two-tailed, unpaired Student's t-tests were used unless otherwise specified. Source data are provided as a Source Data file *$P < 0.05$, **$P < 0.01$, ***$P < 0.001$.

MPs, platelets produce MPs as a stress response. PMPs are the predominant circulating MPs in prothrombotic and inflammatory disorders[41], suggesting the pivotal regulatory role of these MPs in MK development in pathological and stress states.

Jiang et al. showed that only MK-MPs but not PMPs could be taken up by HSPCs and enhance MK differentiation, which is distinct from our studies. However, an insufficient observation time might account for that difference. Fluorescence and differential interference contrast images taken by Jiang's team showed that PMPs could bind to but not be taken up by HSPCs after 3–5 h of coculture[3]. In contrast, we discovered that more than 30% of CD34[+]-HSPCs had internalized the PMPs 24 h after PMP application by FCM. After 72 h of culture, over 80% of the cells had internalized the fluorescently labeled PMPs, and the fluorescence of PMPs in recipient cells could be captured by fluorescence and differential interference contrast microscopy. In addition, PMPs were internalized by three megakaryocytic cell lineages within 24 h. It is possible that primary hematopoietic cells require more time to internalize PMPs. In our study, PMPs were added to the medium at concentrations from 0.5 to 5 PMPs per CD34[+]-HSPCs, and the intermediate concentration (1–2 PMPs per cell) showed an optimal effect on the differentiation of MKs. In Jiang's study, CD34[+] cells were incubated with 10 PMPs per cell[3]. Therefore, we speculated that excessive MPs might lead recipient cells to produce platelets and disintegrate at an early stage.

TPO has long been known as the major growth factor that drives HSCs toward the MK lineage, and its concentration in the circulation can be regulated by the platelet count. In this way, a steady number of platelets is maintained[43]. However, mice lacking this cytokine or its receptor retain the capacity to produce a substantial number of BM megakaryocytes and platelets at 12–15% of normal to prevent overt signs of spontaneous hemorrhage[44,45], indicating that there must be other regulatory mechanisms for maintaining the platelet number that has not yet been discovered. Several cellular growth factors (interleukin-6, interleukin-11, leukemia inhibitory factor (LIF), etc.) and hormones (estrogen, growth hormone and stress hormones, etc.) have been reported to participate in the regulation of MK differentiation and thrombopoiesis[44,46]. However, even simultaneous knockout of the TPO receptor with IL-6, IL-11, or LIF cannot completely block platelet generation[44], which suggests that extra regulators in addition to TPO and inflammatory cytokines play roles in thrombocytosis. In this study, we found that PMPs also have the ability to promote thrombopoiesis by facilitating MK differentiation in a TPO-independent manner, since an antibody against TPO and the TPO receptor did not block MK differentiation induced by PMPs.

Most importantly, we demonstrated the therapeutic potential of PMPs for thrombocytopenia. PMP injection significantly increased the number of megakaryocytic cells and platelets in mice irradiated at a semilethal dose, suggesting a definite therapeutic effect of PMPs. Moreover, there is increasing evidence suggesting that PMPs participate in thrombus formation[40]. In patients with ITP (immune-mediated thrombocytopenia), elevated levels of PMPs have been shown to be protective against bleeding[47]. Therefore, the therapeutic potential of PMPs in thrombocytopenia should be carefully examined in other diseases. Through further studies, PMPs could meet the urgent need for platelets in the clinic and be used in the pharmaceutical industry.

In conclusion, our findings elucidated the specific effect of PMPs on MK development and platelet generation, which might compensate for TPO regulation. We first revealed that both PMPs and miR-1915-3p inside PMPs acted as positive regulators of megakaryopoiesis. _RHOB_ was first elucidated as a target of miR-1915-3p and PMP. Our present study lays the foundation for further optimization of a MK induction system that is useful in clinical transfusion. This study may contribute to the development of therapeutic strategies for platelet diseases.

## Methods

**Mice and treatments**. Six- to eight-week-old male C57BL/6 mice were used in all experiments. C57BL/6 and GFP transgenic C57BL/6 mice were obtained from commercial vendors (Experimental Animal Center of Academy of Military Medical Sciences). Mice were manipulated and housed according to protocols approved by the Institutional Animal Care and Use Committee (IACUC) of the Institute of Health Service and Transfusion Medicine. Ambient temperatures of ~20–24 °C with 50–60% humidity and 12 h dark/12 h light cycle were used. The study was approved by the Ethical Committee of IACUC and the permit number is IACUC-2014-037. The protocols are compliant with specific Ethical Regulations. The included subjects were randomly assigned to all experimental mouse cohorts, and at least six mice were used per time point for each group unless otherwise indicated.

ALI was induced by a single intraperitoneal injection of 5% carbon tetrachloride solution dissolved with corn oil (Sigma, America) at 10 mL per kg bodyweight. Control mice were injected with an equivalent volume of corn oil. Liver and BM samples were harvested at day 2, day 4, and day 8 after blood collection and perfused with phosphate-buffered saline (PBS). Then, ~20 mg liver samples were lysed by TRIzol (Invitrogen, America) for quantitative RT-PCR. Additional liver tissue was embedded in paraffin after fixation in 10% formalin for hematoxylin-eosin staining.

In whole-body irradiation studies, the mice received 6 cGy total body irradiation using a [60]Co irradiator. We injected different doses of exogenous PMPs into mice immediately after irradiation and observed that 100,000 PMPs per mouse was the best dose for megakaryocytic differentiation and platelet production (Supplementary Fig. 9c, d). Finally, 200 μl saline or an equivalent volume of saline containing PMP ($1 \times 10^5$) was injected into the mice via the tail vein within 6 h of irradiation.

**Cells and PMPs**. Human CB samples were harvested through the umbilical cord after the delivery of normal pregnancies with the patient's informed consent.

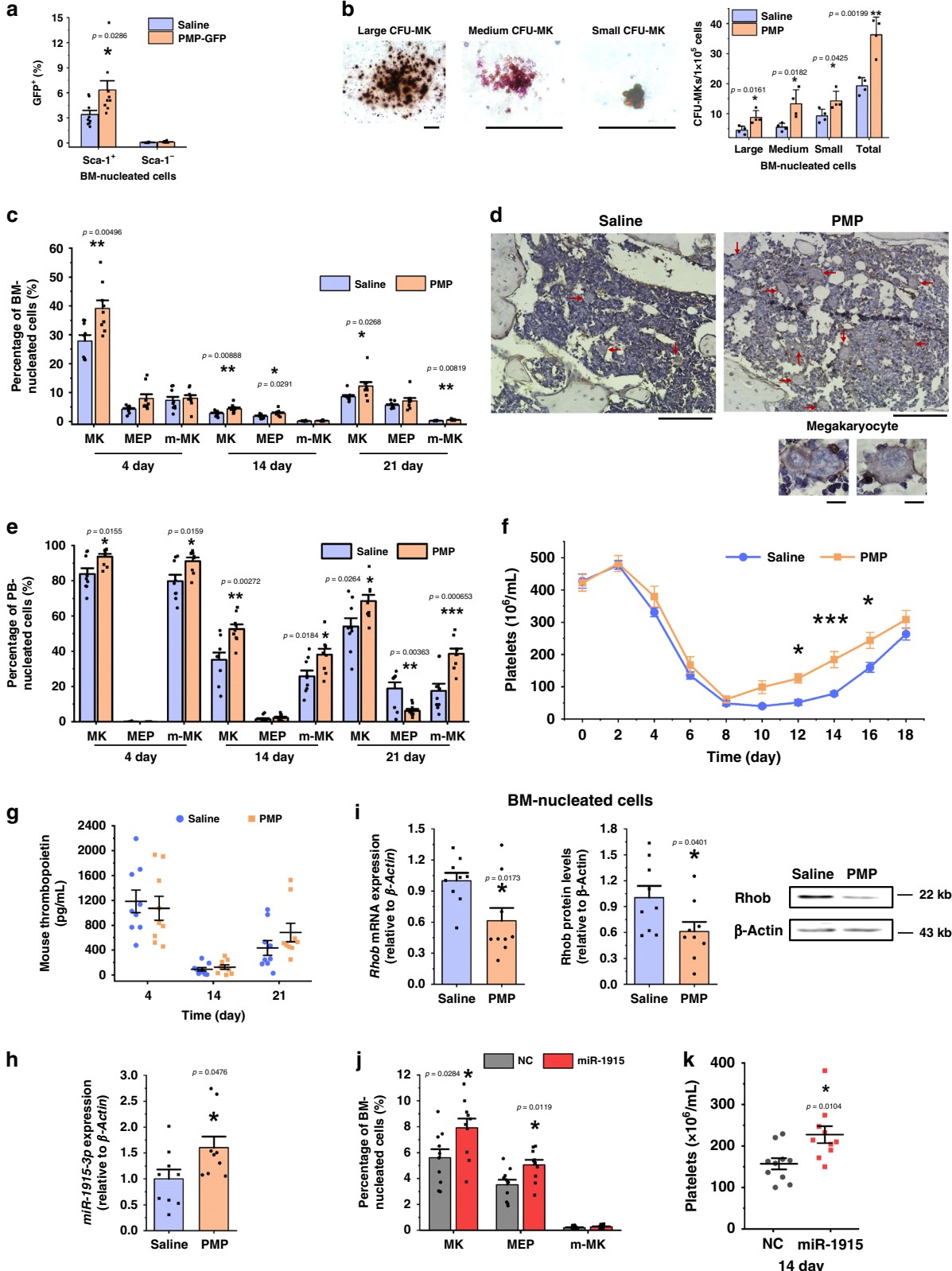

Research has been carried out in accordance with the approval of the ethics committee. The studies involving human subjects were approved by the Ethical Committee of Institute of Health Service and Transfusion Medicine. and the permit number is AF/SC-08/028 and the protocols followed are compliant with specific Ethical Regulations. CB samples were processed within 4 h. MNCs were isolated by Ficoll-Hypaque density gradient centrifugation (1.077 g/L, Tianjing Cord Blood Services Foundation, China). Then, CD34 Microbeads Kit and CD61 Microbeads (MiltenyiBiotec, Germany) were used for the purification of CD34$^+$/CD61$^-$HSPC

from the MNCs before MK differentiation and CD34$^+$/CD61$^+$ CFU-MK cells and CD34$^-$/CD61$^+$ promegakaryocytes (pro-MKs) from the MNCs after MK differentiation in vitro for 3–6 days. The cells were isolated by magnetic force following the manufacturer's instructions[48]. HUVECs were isolated from umbilical cord of healthy volunteers after the delivery of normal pregnancies with the patient's informed consent.

Platelet-derived microparticles (PMPs) were isolated from activated platelets. First, for platelet isolation and purification, human platelets were isolated from the

**Fig. 6 Injection of exogenous PMPs stimulated MK differentiation and platelet production. a** PMPs from the GFP transgenic mice were isolated and transfused into the same strain of C57BL/6 mice. The percentage of GFP$^+$ cells in the Sca-1-positive and Sca-1-negative nucleated cells was analyzed by FCM. **b** PMP treatment increased the CFU-MK formation of the BM nucleated cells. The mice were irradiated and injected with homologous PMPs or saline (control). Three days later, the BM was harvested for megakaryocytic colony-forming assays. The typical morphology of CFU-MK mouse colonies of different sizes is shown on the left (Scale bars: 100 μm). The statistical analysis is shown on the right. ($n = 4$ mice). **c** PMP enhanced the megakaryocytic lineage proportion of the BM nucleated cells. Representative FCM plots are shown in Supplementary Fig. 9c. **d** The femur sections stained with hematoxylin and a CD41 antibody indicate more polyploid and CD41$^+$ cells with PMP injection. The tissue samples were harvested 21 days post injection, and the red arrows indicate large, polyploid and CD41-positive megakaryocytes (Scale bars: 100 μm). Representative fields of megakaryocytes in the BM are shown on the left (Scale bars: 10 μm). **e** PMPs enhanced the megakaryocytic lineage proportion of the PB nucleated cells. Representative FCM plots are shown in Supplementary Fig. 9d. **f** PMPs improved platelet recovery as indicated by the platelet count ($n = 12$ mice, two-way repeated-measures ANOVA with Bonferroni's multiple comparison tests, $p = 0.0171$). **g** Plasma TPO was unaffected after PMP treatment. **h** qPCR indicated that miR-1915-3p was upregulated in the BM nucleated cells 2 days after PMP injection. **i** *Rhob* was downregulated in the mouse BM nucleated cells after PMP injection as evaluated by qPCR (left) and Western blotting (right) (normalized to the mouse after saline injection). Transfusion of PMPs overexpressing miR-1915-3p enhanced the megakaryocytic lineage proportion of the BM nucleated cells **j** and the platelet count in the PB **k** ($n = 10$ mice). Each time point in each animal cohort represents a mean ± SE.M. of 9 mice, *P*-value. Two-tailed, unpaired Student's *t*-tests were used unless otherwise specified. Source data are provided as a Source Data file *$P < 0.05$, **$P < 0.01$, ***$P < 0.001$.

supernatant of umbilical CB after Ficoll-Hypaque density gradient centrifugation and pelleted by centrifugation at $2000 \times g$ for 20 min at RT, and mouse platelets were isolated from the whole blood supernatant, which was collected by retroorbital bleeding, dilution with EDTA buffer and centrifugation at $200 \times g$ for 15 min. The purity of CD41a-positive human or mouse isolated platelets was more than 95% (Supplementary Fig. 1a, c left panel, d left panel). Second, the platelets were resuspended in saline (pH = 7.4) containing 1 mM calcium and 1 U per mL thrombin for 15 min at 37 °C. The PMPs were released into the supernatant. Contaminating remnant platelets were removed by centrifugation at $2000 \times g$ for 20 min at 4 °C and filtered through a 0.8 μm filter unit (Millipore, America). Then, the supernatants containing PMPs were centrifuged at $20,000 \times g$ for 120 min at 4 °C. PMPs were resuspended in saline prelabeled with PKH26, PHK67 (Sigma, America) or fluorescent antibodies and washed when required.

To manipulate miRNA expression in PMPs, miRNA mimics were transfected into platelets, the parents of PMPs. The transfection control was a double-stranded RNA with the sequence 5'- UUCUCCGAACGUGUCACGUTT-3' for the sense strand and 5'- ACGUGACACGUUCGGAGAATT-3' for the antisense strand. The procedure for platelet transfection was based on the protocol by Wei Hong[49]. Platelets ($1 \times 10^8$) were resuspended in modified Tyrode's buffer and transfected with 600 nM miRNA with or without FAM labeling (GenePharma, China) using Lipofectamine 2000 transfection reagent (Invitrogen, America), subjected to gentle constant shaking at room temperature, and harvested at 48 h, if necessary. Subsequently, PMPs were harvested from these platelets as described above.

**Complete blood counts and PMP quantification**. Complete blood counts and hemoglobin quantification were performed using whole blood collected via tail bleeding. A Celltac E fully automated hematology analyzer (Nihon Kohden, Japan) was used for the analysis.

In PMP quantification, mouse plasma was harvested and checked directly. In brief, mouse blood (20 μL) was diluted with EDTA buffer and centrifuged at $2000 \times g$ for 20 min at RT. The supernatant containing PMPs was incubated with anti-CD41 and anti-CD42 (eBiosciences, America) antibodies and quantified by FCM using Truecount tubes equipped with fluorescent count beads (2 μm, BD Biosciences, America). Fluorescent size beads (0.22, 0.45, 0.88, and 1.35 μm, Spherotech, America) and latex beads (1.1 μm, Sigma, America) were used to set a size exclusion gate. PMPs were detected as particles <1 μm in size that stained positive for CD41 and CD42. Data were converted to the number of MPs per 1 μL of whole blood.

Mouse plasma thrombopoietin (TPO) levels were quantified using a Thrombopoietin Mouse ELISA kit (Abcam, America). In brief, $1 \times 10^5$ PMPs were lysed directly with 1% NP40 solution (100 μl) on ice for 30 min. Then, their TPO levels were quantified using a human TPO ELISA kit.

**Antibodies and staining reagents**. The following antibodies from eBioscience were used for FCM: FITC-anti-mouse-CD41, APC-anti-mouse-CD42, PE-anti-mouse-CD71, PE-anti-human-CD34, APC-anti-human-CD61, FITC-anti-mouse-CD34 and PE/FITC- anti-human-CD41a. A rat anti-mouse-CD41 monoclonal antibody (Abcam) diluted 1:50 in PBS was used for immunohistochemical staining. CD63 mouse monoclonal antibody (Santa Cruz Biotechnology), RHOB monoclonal antibody (ABclonal) and GAPDH monoclonal antibody (Cell Signaling Technology) were diluted 1:1000 and used for Western blotting, and 1 μg per mL human TPO antibody or human TPO receptor antibody (R&D Systems) was used to neutralize TPO in PMPs. The information on the antibodies used in our study is provided in Supplementary Table 2.

**Cell flow cytometry**. Mouse BM cells were obtained by flushing and crushing the hind leg BM in PBS. PB cells were collected via tail bleeding. BM or PB nucleated cells were isolated from whole BM or PB after lysing on ice with red blood cell lysis

solution and washed with PBS. Cultured cells were collected and washed twice with PBS or EDTA buffer. Cells were labeled with monoclonal antibodies in PBS for 30 min at 4 °C and washed twice before analysis by a BD FACSCalibur. For DNA content analysis, cells were permeabilized with cold 70% ethanol and stained with propidium iodide (50 μg/mL) with RNase A. Flow cytometry were performed using FACS Diva Software (Version 7, BD Biosciences), FlowJo software (Version 10, TreeStar).

**Quantitative real-time polymerase chain reaction (qPCR)**. We obtained primer sequences from PrimerBank (https://pga.mgh.harvard.edu/primerbank/)[50] and performed RNA isolation, reverse-transcription and a qPCR assay according to the manufacturer's instructions (QIAGEN, German). The primers are listed in Supplementary Table 1. Human gene expression levels were normalized to the housekeeping gene *GAPDH*, and mouse genes were normalized to *Hprt*. Analysis of mature miRNA expression in PMPs or cells treated with PMPs was normalized to miR-4454 or cell number, and other analyses were normalized to U6. qPCR was performed using Bio-Rad iQ5 and Bio-Rad CXF Manager software (Bio-Rad).

**Colony-forming unit (CFU) and CFU-MK assay**. Colony medium and staining kits were purchased from StemCell Technologies. Three days after PMP treatment, 500 human CB-HSPCs were plated in methylcellulose-based medium with recombinant cytokines (H4434) for human hematopoietic cell CFU assays. Another 5000 cells were plated in a collagen-based kit with cytokines (04961) for the human CFU-MK assay. For the ex vivo assay, 3 days after PMP injection, 50,000 mouse BM nucleated cells were plated in methylcellulose-based medium with recombinant cytokines (M3434) for mouse hematopoietic cell CFU assays. Another 500,000 cells were plated in a kit containing collagen and lipids (04974) for the mouse CFU-MK assays. The colony formation units were incubated at 37 °C in a humidified atmosphere and assessed 7 days after plating for CFUs and 10 days after plating for CFU-MKs. Mouse MKs and early megakaryocytic progenitors were stained according to the specific enzymatic reaction of acetylcholinesterase, which resulted in brown granular deposits of copper ferrocyanide in the cytoplasm. And human CFU-MKs were visualized by staining by CD41 positivity (04962). Counterstaining with Harris' hematoxylin showed violet-stained nuclei in all cells. Colonies containing more than 3 MKs were scored as CFU-MKs and subdivided based on their size: small (3–20 cells), medium (21–49) and large (≥50). Cells were obtained from at least four different donors, and each sample was tested with three independent experiments.

**Confocal microscopy and transmission electron microscopy**. Human PMPs stained with PKH67, a green fluorescent dye, or an equivalent volume of saline were added to the culture medium with CB-HSPCs (24–72 h), HUVECs, K562 cells, UT-7 or other cells for 0.5–24 h. Afterward, nonadherent PMPs were removed and washed twice with EDTA buffer. Subsequently, the CB-HSPCs were stained with anti-CD41-PE, and nuclei were counterstained with DAPI. Images of CB-HSPCs and other cells containing PMPs were taken with a confocal microscope (Zeiss LSM 510, German) and quantified microscopically (Nikon Corporation, Tokyo, Japan). For direct PMP imaging, PMPs were stained with phosphotungstic acid, placed directly on carbon-coated copper grids and imaged by TEM (Hitachi H7650, Japan).

**MicroRNA and mRNA expression profiling**. Total RNA extracted from $5 \times 10^6$ PMPs ($n = 5$) was used for the Human miRNA Microarray based on the Agilent GeneChip (OE Biotech, Shanghai, China). Total RNA extracted from the CB-MNCs cultured in the presence or absence of PMPs was used for human gene expression analysis using an Affymetrix GeneChip Microarray (OE Biotech's, Shanghai, China). In brief, $1 \times 10^6$ CB-MNCs from three different donors were

isolated and cultured in megakaryocytic induction medium supplemented with 20 μL of saline or an equivalent volume of saline containing $2 \times 10^6$ PMPs from another two donors. Three days after incubation, cells were washed twice and harvested for further experiments.

Analyses were performed by OE Biotech. miRNA and mRNA expression was log-2 transformed after quantile normalization. miRNAs whose expression was higher than the median value of every sample were determined to be highly expressed miRNAs in PMPs. Differentially expressed mRNAs (>1.2-fold change or <0.833-fold change) in cells with PMPs were selected for further bioinformatics analysis. The expression differences discovered by microarray were confirmed by qPCR with selected miRNAs or mRNAs.

**Plasmid construction**. The 512-bp human pri-miR-1915-3p coding sequence was amplified by PCR from human CB MNC cDNA and subcloned into pcDNA3.0-neomycin plasmids. We constructed miR-1915-3p sponges by synthesizing and tandemly arraying five different predicted miR-1915-3p binding sites (Thermo Fisher Scientific Shanghai, China) and inserted the fragment into the plasmid pcDNA3.0-neomycin. Stable miR-1915-3p-overexpressing leukemia lines were developed using the recombinant plasmid (pcDNA3.0-pri-miR-1915-3p-neomycin), and cells were downregulated using pcDNA3.0-miR-1915-3p-sponge-neomycin. The empty vector (pcDNA3.0-neomycin), carrying the neomycin resistance gene as a selection marker, was used as a transfection control.

Three databases (TargetScan, microRNA.org, and PITA) were uesd to predicted the target genes of miRNA. Links to databases TargetScan, microRNA.org, and PITA are http://www.targetscan.org/, http://www.mirbase.org/, http://www.mirbase.org and http://www.pictar.org/, respectively.

The 433-bp wild-type *RHOB* 3′ UTR fragment (383–815) containing the predicted miR-1915-3p binding sites was generated by amplifying the segment from human CB MNC cDNA. Xba I sites were incorporated into primer sequences, and Xba I-digested PCR products were ligated into the Xba I site of the pGL3 control vector (Promega, Madison, WI, USA). For the *RHOB* 3′ UTR mutation, primers were used to amplify both flanks of the putative binding site (607–629) and resulted in two fragments: (Xba I)-Left Fragment (383–606)-(blunt end) and (blunt end)-Right Fragment (630–815)-(Xba I). The 2 fragments were blunt-end ligated and cloned into the Xba I-digested pGL3 vector. Primers were designed as follows: wild-type *RHOB* sense 5′-TCTAGAATCAGATGTTCGCCCTTCACC-3′, wild-type *RHOB* antisense 5′-TCTAGACATCAGTTTCTTGGACTGAAC-3′; mutation *RHOB* sense 5′-AAGACATTTGCAACTGACTT-3′, and mutation *RHOB* antisense 5′-AAGTCCTGTTCATGCTTG-3′.

**Dual-luciferase reporter assay**. K562 cells ($2 \times 10^5$) were cotransfected with 800 ng pGL3-wild-type-RHOB (or pGL3-mutation-RHOB or pGL3 control vector) and 50 nM miR-1915-3p mimic (or negative control mimics) plus 2 ng of the pRL Renilla luciferase vector (Promega) with Lipofectamine 2000 (Invitrogen). Luciferase activity was assayed 48 h after transfection according to the manufacturer's protocol.

**Statistics and reproducibility**. Statistical analysis and results from micrographs were conducted on data from three or more biologically independent experimental replicates. All data are shown as the mean ± S.D. unless otherwise indicated. Selective statistical analysis was conducted for each experiment using GraphPad Prism 5 (GraphPad software, Inc.). $*p < 0.05$; $**p < 0.01$; $***p < 0.001$ (two-tailed $t$ test).

**Reporting summary**. Further information on research design is available in the Nature Research Reporting Summary linked to this article.

## Data availability

Source data are provided with this paper. The source data underlying Figs. 1a, c–j, 2b–i, 3c–f, 4a–i, 5c–h, 6a–c and 6e–k and Supplementary Figs. 1c–f, h, 2c, d, 3b, d, 4b–d, 5a, b, 6a, b, d–f, 9c–e, h and 10b–d are provided as a Source Data file. All novel microarray data were deposited to the Gene Expression Omnibus (https://www.ncbi.nlm.nih.gov/geo/) under accession number GSE152078 and GSE152079. The authors declare that all data supporting the findings of this study are available within the article and its Supplementary Information files.

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

## Acknowledgements

This work was supported by grants from the Beijing Natural Science Foundation (No. 7194303), National Key Research and Development Program of China (No. 2017YFA0103100, No. 2017YFA0103103, No. 2017YFA0103104), Guangzhou Health Care and Cooperative Innovation Major Project (No. 201803040005), Science and Technology Program of Guangzhou (No. 201604020007), Nature Science Foundation of China (No. 81800103, No. 81872553).

## Author contributions

M.Q. performed the experiments and collected the data with assistance from X.Z. (histopathology and animal experiments), F.F. (cell isolation and culture and animal experiments), S.W. (qRT-PCR and animal experiments), L.X. (animal experiments and solution preparation), Z.F. (FACS and PMP characterization), X.X. conceptualized the project, obtained funding, trained the students, designed, and set up the experiments, analyzed the data, and wrote the manuscript with help from Q.Z. (resources and training), L.C. (training and consultation), W.Y. (funding and training), X.P. (funding and writing).

## Competing interests

The authors declare no competing interests.
