## [Peer Review File · Nature Communications]

Reviewers' comments:

Reviewer #1 (Remarks to the Author):

The authors made some intriguing observations regarding the potential role of platelet-derived microparticles in MK differentiation and platelet biogenesis. While the hypothesis exciting, numerous concerns were raised.

Text in result sections mentions that necrosis and granulomas became apparent by Day 4 and increased on Day 8. The figure (Fig. S1) does not show this data.

While thrombocytosis is present from Day 8 through 12, whether other cellular lineages are also modulated is not presented. To further support the role of PMP in MK production and platelet biogenesis, it is important to determine whether other cells are also affected. What is the impact of ALI on RBC, neutrophils, monocytes and lymphocytes in blood for instance?

Data in figure 2C are not convincing; I have a hard time to believe there is any significant difference between CON and ALI conditions. The statement in result section "marker of MK was more highly expressed in bone marrow in ALI" (line 95) is an overstatement. Moreover, these data are calculated with percentage, a small change in proportion of other lineages could impact the apparent proportion of MK, especially when considering the very low (if any) difference between each group.

More information on key differences between CON and ALI are necessary: it is mentioned that mature MK (CD41/CD61+), and late stage MK are more numerous but these data are not presented.

The authors mention that platelets can be released from reserve in the spleen but they do not investigate MK in spleen in ALI. PMP could have access to MK in spleen, and at this stage it is not clear why this is overlooked. The authors should provide details regarding MK in spleen as well as in bone marrow.

How PMP were quantified is unclear. As many key conclusions are based on these quantifications, it sheds further doubts on the findings that the authors report. For instance, it is mentioned that PMP are pelleted prior quantification, which goes against state-of-the-art methodologies in PMP research (Linares JEV 2015). Moreover, the authors are using dyes PKH26 or PKH67, which too are known to be unsuitable to quantify PMP in biological specimen (Puzar BBA 2018). How exactly flow cytometry provide quantitative enumeration of PMP (number/ul), and how PMP are detected and distinguished from larger debris, is omitted in methods. What are the flow cytometry parameters? In short, none of the recognized minimal requirement for PMP quantification, as established by the International Society of Extracellular Vesicles, are respected.

The authors have data that suggest that TPO may not be involved in MK differentiation and platelet production in ALO, and use it to support the role of miR's, but they fail to determine whether other relevant molecules, such as inflammatory cytokines (e.g. IL1, IL6), are also elevated and whether they contribute. Many other players could increase MK in ALI model.

How is it useful to determine whether PMP are internalized in HUVEC? Moreover, it is already reported by different groups that indeed endothelial cells internalize PMP. Use of PKH family dye is not appropriate for these internalization assays (Puzar BBA 2018).

That PMP increases CD41 expression by MNC is interesting. However, the increase is immediate, and observed by T=0, possibly due to the association of PMP with MNC. Protein synthesis inhibition should be utilized in these assays to assess to actual expression of CD41 by MNC vs the association of PMP to MNC. Transfection of irrelevant miR, rather than empty vector, is needed in

control experiments.

MK cell lines do not undergo hematopoiesis in any physiologically meaningful way. All experiments done with these need to be repeated in primary cells.

miRNAs were detected from healthy conditions, yet the model of thrombocytosis was in a disease state. Won't the platelet and PMP components be very different in ALI?

Even a 200-fold overexpression of miR-1915-3p barely suffices to promote MK colony formation unit. In Fig. 4B, C the authors observe apparent statistically significant increases, however whether the increase is biologically significant is doubtful given the almost imperceptible difference between each group.

The RHOB expression is potentially reduced by PMP treatment (Fig 5d) but there is no stats presented.

The transfusion of PMP in irradiated mice is an interesting approach. However, that PMP transfusion actually enhanced the recovery of platelets is not obvious from Fig 9 d,e. The concentration of PMP injected in each mouse (only 100 000) is surprisingly low, and it is unclear how this concentration was chosen, and how this low number can actually play a role above the contribution of the endogenous PMP present in irradiated mice.

Introduction

"The activation and consumption of platelets during thrombocytosis will result in releasing PMPs in a large quantity ». What does it mean, is it really demonstrated?

The introduction is also under-cited

Megakaryocytosis is not a word

Spleen MKs/platelets are thought to only contribute to platelet count in mice. There is no evidence that this is also true in humans.

English has to be improved, and typos/errors are present in text and many figures (e.g liver injure, presentage).

Reviewer #2 (Remarks to the Author):

This is a very high quality manuscript that is well written, very novel, strong data, strong-designed experiments, and is very suitable for Nature Communications. I highly recommend publication. It is really solid with a great mix of in vitro and in vivo and clinically relevant data. It clearly moves the field forward and I only have one recommendation.

The authors model predicts that the activated microvesicles go into the bone marrow where they may interact with HSC/ MKs. Can they test this by labeling the microvesicles and looking in the marrow of the injected mice and or seeing if adding microvesicles to to their cultures later (when there is megakaryocytic cells) stimulate platelet production in vitro?

Reviewer #3 (Remarks to the Author):

Summary

Qu and colleagues present a study that assessed platelet derived microparticles (PMPs) and their role in megakaryocyte differentiation and platelet production. To assess the function of PMPs, human and mouse PMPs were utilised, they also incorporated in vivo and in vitro studies using a mouse carbon tetrachloride (CCl4) toxicity model to emulate human acute liver injury (ALI), three megakaryocytic cell lines and an irradiated mouse model to validate the role of PMPs in vivo. They demonstrate that in response to ALI, PMPs are elevated which coincides with megakaryocyte differentiation, in parallel, TPO levels were unchanged, therefore not playing a role in this mechanism. Using three megakaryocyte cell lines, PMP internalisation promoted MK differentiation and maturation in vitro. The most elaborate part of the study was the identification of mir-1915-3p as a regulator of megakaryocyte differentiation which functioned through suppressing Rho GTPase family member B (RHOB) expression. This study paves the way for future work to delineate therapeutic strategies in order to replenish the platelet population in platelet associated diseases.

Overall the data presented in the study supports the idea that PMPs can modulate megakaryocyte differentiation, however further experiments need to be carried out to confirm the proposed mechanism which is a major caveat of the manuscript.

Major comments:

- There is a disconnect between the in vivo data and in vitro data. While some of the data generated in vitro is interesting, it is unclear how these results mechanistically relate to results from in vivo experiments in fig 1 and 6.
- Figure 6 E; This result is not convincing. I am not convinced that there is a statistical significance between the two groups. The platelet count should be shown as units per ml or μl , not a percentage.
- FIG 1. Fig 1A. I suggest including actual platelet counts. Not sure what units are used at the moment
- 1C. Is the mature meg population changing? I suggest plotting the CD41/CD42 population separated from the progenitor cells.
- 1E. Serum TPO levels. Unchanged in treated mice, however described in text as reduced. Unclear why levels increased in control mice?
- I appreciate the experiments that Qu and colleagues carried out, however the authors did not assess RHOB and miR-1915-3p expression to confirm their role in modulating PMP function and megakaryocyte differentiation following the in vivo experiments. As the authors confirmed that decreasing miR-1915-3p expression attenuated CD41+/CD61+ populations in the cell lines (fig 4), key experiments would be to assess RHOB protein and gene expression in megakaryocytes derived from the irradiated mouse model untreated or treated with PMPs. RHOB activity should also be assessed in purified megakaryocytes from this model. miR-1915-3p expression should also be analysed in megakaryocytes in the mouse models following PMP injection.
- Similarly, the above-mentioned experiments should be carried out in the ALI mouse model that demonstrated thrombocytosis linked to elevated PMP formation and megakaryocyte differentiation and maturation. The authors need to demonstrate that RHOB gene and protein expression is attenuated in this model and possibly RHOB activity.
- FIG 2. Where is the ploidy data?
- FIG3. The changes observed in D are small and the trend look the same for all miRs included and not specific to miR-1915-3 p.
- Row 72 "and release the mechanism underlying"

What does this mean?

- Can PMPs be pre-treated with an RNase prior to injections? if it is impossible to target miR-1915-3p expression then this should be considered and should be included as a control when treating the irradiated or ALI mice to validate at least in part, that a micro RNA has a role in this mechanism.
- Overall there is an inconsistent size of graphs and font in figures and throughout the text. Additionally, grammatical errors are littered throughout the article that need to be addressed.

Minor comments:

- Row 46 "thrombocytosis" should be replaced with thrombosis.
- Figure 1A: no units are labelled on the y-axis.
- Figure 1B: Acute liver injury should be labelled.
- Figure 1C: Percentage of CD41+ cells on Y axis.
- Figure 1G: In the control vs ALI mouse model, the expression of PMPs was assessed over 6 days. The authors stained for CD41, and CD62 to determine the microparticles from activated platelets, however CD62+ needs to be labelled on fig 1G as only CD41+ MPs may refer to megakaryocyte MPs.
- Figure 1 legend; (a) During CCl4 induced acute liver injury (ALI).
- FIG 1 legend. Progenitor cells are described as "neonatal" but they have used adult mice?
- Figure 2: Image quality/resolution is poor
- Figure 3 A; heat map needs to be revised, the labels are illegible, perhaps not all the miRNAs need to be shown.
- Figure 4: legend title- Altering the expression of miR-1915-3P effects MK differentiation and maturation
- Figure 5 C, F, G: y axis should state gene expression relative to Gapdh
- Supplementary data:
 - Figure S1B: haemoglobin units are not stated
 - Figure S1C: The percentage for double positive CD42/CD71+ population should be included.
 - Figure S7 is missing.
 - Figure S8: A description of the schematic figure needs to be included describing the proposed mechanism of megakaryocyte differentiation independent of TPO.

Answers to the reviewers

We would like to express our sincere thanks to the reviewers for the constructive and positive comments.

Replies to reviewer 1

Q: The authors made some intriguing observations regarding the potential role of platelet-derived microparticles in MK differentiation and platelet biogenesis. While the hypothesis exciting, numerous concerns were raised.

Text in result sections mentions that necrosis and granulomas became apparent by Day 4 and increased on Day 8. The figure (Fig. S1) does not show this data.

A: By consulting the relevant literatures (Allman MJ, Gastroenterol Hepatol, 2010), we found that CCl₄ injection is commonly used to establish an acute liver injury model, and the injury usually begins at day 2. Hepatic inflammation and proliferation could be assessed by immunostaining for hematoxylin–eosin and Ki67, respectively. We have photomicrographs of the liver 2 days after CCl₄ administration. For improved clarity, we have the destroyed structure indicated with white dashes. Representative photomicrographs of liver immunostaining for Ki67 are added as well in the new submission (Supplemental Figure S1a and b). As suggested by the reviewer, we noted that centrilobular necroinflammation and regeneration peaked at day 2 and was nearly resolved by day 8.

Q: While thrombocytosis is present from Day 8 through 12, whether other cellular lineages are also modulated is not presented. To further support the role of PMP in MK production and platelet biogenesis, it is important to determine whether other cells are also affected. What is the impact of ALI on RBC, neutrophils, monocytes and lymphocytes in blood for instance?

A: In our previous submission, we demonstrated that the hemoglobin content was consistent in the ALI group during the detection period, indicating the consistent the blood volume. During the detection period, the RBCs were relatively consistent as well and only showed a decline at day 10 post ALI. The impact of ALI on other blood cells was also assessed and is shown in the new submission. Inflammation from ALI caused the upregulation of immune cells, including neutrophils, monocytes and lymphocytes. We have shown these alterations in the new submission (Supplemental Figure S1c). Accordingly, the increase in platelets after ALI is one of the most profound and interesting phenomena.

Q: Data in figure 2C are not convincing; I have a hard time to believe there is any significant difference between CON and ALI conditions. The statement in result section “marker of MK was more highly expressed in bone marrow in ALI” (line 95) is an overstatement. Moreover, these data are calculated with percentage, a small change in proportion of other lineages could impact the apparent proportion of MK, especially when considering the very low (if any) difference between each group.

A: We apologize for the mistake we made in Figure 1c. We labeled the plots for day 4 as the plots for day 8. This error may have resulted in an incorrect interpretation of our results by the reviewers. We thank the reviewer for his/her care in reviewing the manuscript, and a correct figure is attached below.

We agree with the reviewer that the change in proportion might be due to changes in other lineages. To improve the results, we have directly shown the relative bone marrow megakaryocyte counts in the new submission (Supplemental Figure S1h). We still observed a significant improvement when comparing the megakaryocyte numbers between the ALI and control groups. The immunohistochemistry analysis confirmed the increased megakaryocytes in the ALI group as well.

Q: More information on key differences between CON and ALI are necessary: it is mentioned that mature MK (CD41/CD61+), and late stage MK are more numerous but these data are not presented.

A: On day 8 of the CCl₄ injection, the percentage of CD41⁺ cells, which indicates all MKs in the BM, was 22.76% vs. 28.47% for the CON and ALI groups, respectively (p=0.000293); the percentage of CD41⁺/CD71⁺ cells (megakaryocyte-erythroid progenitors, MEP) was 15.14% vs. 18.26% for the CON and ALI groups, respectively (p=0.004458); and the percentage of CD41⁺/CD42⁺ cells (mature megakaryocyte, m-MKs) was 2.23% vs. 3.14% for the CON and ALI groups, respectively (p=0.000179). These data are presented in Figure 1c of the new submission.

Furthermore, the difference was also shown in the ploidy levels, which represent the maturation of MKs. We assessed the differences in BM cell ploidy and showed the results in Figure 1d.

Q: The authors mention that platelets can be released from reserve in the spleen but they do not investigate MK in spleen in ALI. PMP could have access to MK in spleen, and at this stage it is not clear why this is overlooked. The authors should provide details regarding MK in spleen as well as in bone marrow.

A: Studies from Lee et al. (Am J Hematol. 1985) suggested that the spleen is a reservoir of platelets. This finding suggested that the increase in platelets after ALI might be due to the release of preserved platelets from the spleen. As requested by the reviewer, we detected the percentage of MKs and the number of platelets in the spleen. The number of platelets in the spleen was also increased significantly, but the percentage of CD41⁺ megakaryocytic cells in the spleen did not show a significant alteration on day 8 in the ALI group (Supplemental Figure S1e and f). The increase in platelets inside the spleen suggested elevated platelet production instead of the release of preserved platelets from the spleen.

Q: How PMP were quantified is unclear. As many key conclusions are based on these quantifications, it sheds further doubts on the findings that the authors report. For instance, it is mentioned that PMP are pelleted prior quantification, which goes against state-of-the-art methodologies in PMP research (Linares JEV 2015). Moreover, the authors are using dyes PKH26 or PKH67, which too are known to be unsuitable to quantify PMP in biological specimen (Puzar BBA 2018). How exactly flow cytometry provide quantitative enumeration of PMP (number/ul), and how PMP are detected and distinguished from larger debris, is omitted in methods. What are the flow cytometry parameters? In short, none of the recognized minimal requirement for PMP quantification, as established by the International Society of Extracellular Vesicles, are respected.

A: We apologize for the unclear description of the PMP quantification.

To determine the levels of PMPs in the ALI mice, we harvested the plasma and analyzed it directly using flow cytometry without centrifugation. The PMPs were first gated with forward scatter and side scatter and then assessed by CD41 and CD62p surface marker expression. The quantification of PMPs was performed via flow cytometry using Truecount tubes equipped with fluorescent count beads (2 μm , BD Biosciences, America). Fluorescent beads of different sizes (0.22, 0.45, 0.88 and 1.35 μm , Spherotech, America) and latex beads (1.1 μm , Sigma, America) were used to set a size exclusion gate.

For PMP isolation and application, briefly, the platelets were first purified from plasma by $2000 \text{ g} \times 20 \text{ min}$ centrifugation. Then, the platelets resuspended in saline (pH=7.4) were treated with 1 mM calcium and 1 U/mL thrombin for 15 min at 37°C . The PMPs were released into the supernatant. Contaminating remnant platelets were removed by centrifugation at $2,000 \times \text{g}$ for 20 min at 4°C and filtered through a 0.8 μm filter unit (Millipore, America). Then, the supernatants containing PMPs were centrifuged at $20,000 \times \text{g}$ for 120 min at 4°C for enrichment. The PMPs were resuspended in saline for further experiments. Centrifugation could not be eliminated in our experiment. In the reference mentioned by the reviewer (Linares JEV 2015), high-speed centrifugation resulted in the aggregation of EVs, which could cause erroneous conclusions about EV composition or phenotype. We determined the origin of the purified microparticles (platelets); as a consequence, there would not be a mixture of PMP contents and phenotype. According to the minimal requirement for PMP quantification (Lotvall et al., JEV 2017), we showed the purity of the platelets that were used for the generation of PMPs in the new submission. The purity of CD41a-positive platelets isolated from humans or mice was more than 95% (Supplemental Figure S1a, c left panel, d left panel). Fluorescent dye labeling was only performed when the uptake of PMPs was traced in the target cells. For this aim, PKH26 or PHK67 (Sigma, America) was added to the suspension of PMPs at 2 $\mu\text{mol/mL}$ for 5 min and washed twice before being supplemented with cell culture medium. The concentration of fluorescent dye we used was much lower than that

reported by Puzar et al. (8 $\mu\text{mol/mL}$), and the centrifugation speed for PMP washing was 20,000 g, much lower than the 100,000 g used by Puzar et al. We consider our data reliable under such circumstances. However, we found that the uptake of PKH26-PMPs was much slower than PKH26 staining in HSPCs (3 days vs. 5 minutes), which suggested a difference in the fluorescence-labeled PMPs and the fluorescent dye. To confirm the difference in PMPs and aggregated fluorescent dyes, we directly generated aggregated fluorescent dye through the same centrifugation method used for the PMPs and showed that less than 5% of the target cells were taken up in the new submission (Supplemental Figure S4b).

Q: The authors have data that suggest that TPO may not be involved in MK differentiation and platelet production in ALI, and use it to support the role of miR's, but they fail to determine whether other relevant molecules, such as inflammatory cytokines (e.g. IL1, IL6), are also elevated and whether they contribute. Many other players could increase MK in ALI model.

A: TPO is a key factor for megakaryopoiesis, which is why we examined the impact of TPO during ALI. The exclusion of the involvement of TPO in megakaryopoiesis suggests that other regulators may be involved. We also learned from Gainsford et al. (Blood 2000) that platelet generation cannot be completely inhibited by simultaneously knocking out the TPO receptor with IL-6, IL-11, or LIF. Accordingly, there are still some unknown factors in addition to inflammatory cytokines that are involved in the regulation of platelet generation. In our study, the significant difference in circulating PMPs after ALI suggested their role in megakaryopoiesis. Further experiments confirmed the role of the PMPs. However, we still cannot exclude the involvement of other factors. We will discuss these issues in the new submission.

Q: How is it useful to determine whether PMP are internalized in HUVEC? Moreover, it is already reported by different groups that indeed endothelial cells internalize PMP.

Use of PKH family dye is not appropriate for these internalization assays (Puzar BBA 2018).

A: PMP internalization by HUVECs is just set in the supplemental materials instead of the main text. Since PMPs are commonly reported to be internalized by endothelial cells, we would like to show these data to confirm the biological characteristics of the PMPs.

Q: That PMP increases CD41 expression by MNC is interesting. However, the increase is immediate, and observed by T=0, possibly due to the association of PMP with MNC. Protein synthesis inhibition should be utilized in these assays to assess to actual expression of CD41 by MNC vs the association of PMP to MNC. Transfection of irrelevant miR, rather than empty vector, is needed in control experiments.

MK cell lines do not undergo hematopoiesis in any physiologically meaningful way. All experiments done with these need to be repeated in primary cells.

A: The reviewer might have misunderstood our results. Please review Figure 2c for the impact of PMPs on MNC differentiation. The first time point is day 5; that is, the effect of PMPs was observed by T=5. In the first 24 hours after PMPs were added, no major difference in surface marker expression was detected in our experiment. This discovery suggested that the difference in surface marker expression might not be because of the direct association of PMPs with MNCs. In addition, we thank the reviewer for the suggestion that cycloheximide (CHX), a protein synthesis inhibitor, should be used to further confirm the expression of CD41 by MNCs upon PMP incubation. In the presence of CHX, PMP supplementation did not result in a significant alteration in the percentage of the CD61⁺ and CD41⁺ populations (Figure 2d). Accordingly, the effect of PMPs must be based on protein synthesis.

For the transfection control, we used irrelevant miRNA for primary cell transfection. The control is a double-stranded RNA with the sequence 5'-UUCUCCGAACGUGUCACGUTT-3' for the sense strand and 5'-ACGUGACACGUUCGGAGAATT-3' for the antisense strand. The empty vector was only used in the cell line transfection. All experiments on the impact of PMPs on

megakaryocyte differentiation have been repeated in primary cells as well. We will emphasize this point in the new submission.

Q: miRNAs were detected from healthy conditions, yet the model of thrombocytosis was in a disease state. Won't the platelet and PMP components be very different in ALI?

A: To our knowledge, the miRNA contents of exosomes vary in different states in physiological conditions, but there is no report discussing the differences in the contents of the PMPs between healthy conditions and disease states. Considering that PMPs are harvested from irritated platelets, we would like to directly connect PMPs to the pathological state [Curr Opin Hematol 2015]. To address the concerns of the reviewer, we performed comparisons of the platelet and PMP miRNA contents between healthy and disease states using the miRNAs discovered in healthy conditions. We found that the miRNAs in the ALI mouse platelets were consistent with those in the normal mice, and in addition to miR-1915-3p (1.65-fold change) and miR-3960 (1.75-fold change), other detected miRNAs were stable in the PMPs of the ALI mice compared to normal controls (Figure 3e). The increase in PMPs during the pathological state (3.26-fold augmentation at day 2 after ALI) might be the key event in the response to this condition.

Q: Even a 200-fold overexpression of miR-1915-3p barely suffices to promote MK colony formation unit. In Fig. 4B, C the authors observe apparent statistically significant increases, however whether the increase is biologically significant is doubtful given the almost imperceptible difference between each group.

A: The unclear description of the study method might have resulted in the confusion of the reviewer. In fact, the primary cells used in the study were obtained from unique cord blood donors, and each sample was tested with 3 independent experiments for data presentation. Because of the variable gene expression between individuals, the error bars show high standard deviations. Therefore, paired-samples t-test analysis was used to confirm the statistical significance in this experiment. The total

CFU-MKs increased to 189/5000 CD34⁺ cells with miR-1915-3p overexpression compared to 163/5000 CD34⁺ cells for the control group. The difference is significant ($p=0.002014$). However, the transfection of miR-1915-3p mimics could only be performed before the plating of the CD34⁺ cells, which limited the effect of the miRNA to the initiated time point only. The individual differences resulted in a large error bar for each group. We have rearranged the plots for improved clarity.

Q: The RHOB expression is potentially reduced by PMP treatment (Fig 5d) but there is no stats presented.

A: We have shown improved pictures of the Western blotting and added the statistical analysis of the grayscale images in the new submission (Figure 5d, f-g).

Q: The transfusion of PMP in irradiated mice is an interesting approach. However, that PMP transfusion actually enhanced the recovery of platelets is not obvious from Fig 6 d,e. The concentration of PMP injected in each mouse (only 100 000) is surprisingly low, and it is unclear how this concentration was chosen, and how this low number can actually play a role above the contribution of the endogenous PMP present in irradiated mice.

A: We compared different dosages of PMPs for the optimal transfusion effect. In our experiment, a higher PMP concentration did not result in better platelet recovery. We found that 100,000 PMPs per mouse was the optimal dosage. We apologize for not describing the detailed procedure for the PMP dosage optimization, which has been added in the new submission (Supplemental Figure S9c-d).

In Fig 6c and 6e, we tried to show which subpopulation of megakaryocytes showed the strongest response to PMPs. The arrangement of the bar plots might result in confusion. There was a statistically significant difference between the PMP and control groups. The recovery of circulating platelets in the PMP transfusion group was faster than that in the control group (Figure 6f, Supplemental Figure S9h). The individual difference might obscure the group difference; however, it is still statistically significant. The p values are attached below.

Table 1. *P* values in figure 6.

Time (day)	Bone marrow (Figure 6c)			Peripheral blood (Figure 6e)		
	4	14	21	4	14	21
MK	0.004962	0.008878	0.02681	0.015467	0.002715	0.026387
m-MK	0.639461	0.255691	0.008192	0.015892	0.018433	0.000653
MEP	0.022743	0.029092	0.152	0.086754	0.446805	0.00363

Time (day)	10	12	14
Platelets (Figure 6f)	0.040486	0.010154	0.012288

Based on the work of Jiang et al. (Control Release 2018), Mk-MPs can target hematopoietic stem/progenitor cells and direct the cells toward the fate of megakaryocytes. PMPs might have a similar effect for Mk-MPs. The disease circumstances might enhance the targeting and effect of PMPs. This factor might be why a small number of PMPs is enough to trigger the generation of platelets in vivo. To further address this issue, we isolated PMPs from the GFP transgenic mice and transfused these GFP-PMPs to trace the homing and uptake of PMPs. The level of GFP was substantially elevated in Sca-1 (stem cell antigen-1)-positive BM-nucleated cells 4 days after GFP-PMP injection, which indicates that PMPs went into the bone marrow and were internalized by Sca-1-positive nucleated cells (Figure 6a, Supplemental Figure S9a). This finding suggested that in irradiated mice, PMPs were primarily taken up by hematopoietic stem/progenitor cells (Sca-1⁺ cells).

Introduction

“The activation and consumption of platelets during thrombocytosis will result in releasing PMPs in a large quantity ». What does it mean, is it really demonstrated?

The introduction is also under-cited

A: A review article (Curr Opin Hematol 2015) summarized the generation of PMPs. As noted by the reviewers, the activation of platelets by physiological agonists such as collagen, thrombin, the complement membrane attack complex C5b-9, lipopolysaccharide, immune complexes and viruses triggers the release of microparticles. During injury, endothelial cells send out messages and recruit platelets to the site of injury. A platelet activation cascade occurs for platelet adhesion, aggregation and clot retraction. These findings explain the sentence the reviewer noted. We have added the reference in the new submission.

Q: Megakaryocytosis is not a word

A: We adopted the word megakaryocytosis from Ng et al. (Proc Natl Acad Sci U S A. 2014) and Yang et al. (Clin Case Rep. 2015). The word indicates the generation of megakaryocytes at high levels.

Q: Spleen MKs/platelets are thought to only contribute to platelet count in mice. The is no evidence that this is also true in humans.

A: We agree with the reviewer's point. However, it is difficult to determine the response of the spleen to ALI in human patients. We can only obtain the in vivo data from mice. Confirming the positive effect of PMPs on bone marrow megakaryogenesis is sufficient for the functional study of PMPs. In addition, we determined the number of megakaryocytes and platelets in the mouse spleen under ALI to further confirm the only contribution to the platelet count of the spleen MKs/platelets. The data were added in the new submission (Supplemental Figure S1e and f).

Q: English has to be improved, and typos/errors are present in text and many figures (e.g liver injure, presentage).

A: Thank you very much for noting the sentence structure and grammatical issues in our manuscript. According to the comments from you and the editors, we polished the manuscript with professional assistance from an English editing company.

Replies to reviewer 2

This is a very high quality manuscript that is well written, very novel, strong data, strong-designed experiments, and is very suitable for Nature Communications. I highly recommend publication. It is really solid with a great mix of in vitro and in vivo and clinically relevant data. It clearly moves the field forward and I only have one recommendation.

Q: The authors model predicts that the activated microvesicles e. Can they test this by labeling the microvesicles and looking in the marrow of the injected mice and or seeing if adding microvesicles to to their cultures later (when there is megakaryocytic cells) stimulate platelet production in vitro?

A: We thank the reviewer for the appreciation and recommendation. We isolated PMPs from the GFP transgenic mice and transfused these GFP-PMPs to trace the homing and uptake of PMPs. The expression of GFP was substantially elevated in Sca-1 (stem cell antigen-1)-positive BM-nucleated cells 4 days after GFP-PMP injection, which indicates that the PMPs went into the bone marrow and were internalized by Sca-1-positive nucleated cells (Figure 6a, Supplemental Figure S9a).

Based on our previous studies, a direct positive effect of PMPs on megakaryocyte platelet generation is possible. We performed an in vitro experiment with PMPs added to the megakaryocyte culture media to further demonstrate that PMPs can stimulate platelet production. From the same number of seeded pro-MKs, we obtained more PLTs in the pro-MKs treated with PMPs (0.1323 ± 0.03569 PLTs per MK for saline versus 2.9487 ± 0.6957 PLTs per MK for PMPs) (Figure 2h, Supplemental Figure S3f).

Replies to reviewer 3

Summary

Qu and colleagues present a study that assessed platelet derived microparticles (PMPs) and their role in megakaryocyte differentiation and platelet production. To assess the function of PMPs, human and mouse PMPs were utilised, they also incorporated in vivo and in vitro studies using a mouse carbon tetrachloride (CCl4) toxicity model to emulate human acute liver injury (ALI), three megakaryocytic cell lines and an irradiated mouse model to validate the role of PMPs in vivo. They demonstrate that in response to ALI, PMPs are elevated which coincides with megakaryocyte differentiation, in parallel, TPO levels were unchanged, therefore not playing a role in this mechanism. Using three megakaryocyte cell lines, PMP internalisation promoted MK differentiation and maturation in vitro. The most elaborate part of the study was the identification of mir-1915-3p as a regulator of megakaryocyte differentiation which functioned through suppressing Rho GTPase family member B (RHOB) expression. This study paves the way for future work to delineate therapeutic strategies in order to replenish the platelet population in platelet associated diseases. Overall the data presented in the study supports the idea that PMPs can modulate megakaryocyte differentiation, however further experiments need to be carried out to confirm the proposed mechanism which is a major caveat of the manuscript.

Major comments:

Q: There is a disconnect between the in vivo data and in vitro data. While some of the data generated in vitro is interesting, it is unclear how these results mechanistically relate to results from in vivo experiments in fig 1 and 6.

A: We thank the reviewer for his/her consideration. The in vivo study first suggested the correlation between PMP upregulation and megakaryocytogenesis using the ALI model (Figure 1). Subsequently, we performed in vitro studies to confirm the positive effect of PMPs on megakaryocyte development and platelet generation. The underlying mechanism was explored in vitro with primary cells and leukemia cell lines. Finally, we injected PMPs into mice with thrombocytopenia to verify the

Time (day)	10	12	14
Platelets (Figure 6f)	0.040486	0.010154	0.012288

Q: FIG 1. Fig 1A. I suggest including actual platelet counts. Not sure what units are used at the moment

A: Similar to the question raised before, the platelet counts are shown as relative levels. We have shown the raw data in Supplemental Figure S1d.

Q: 1C. Is the mature meg population changing? I suggest plotting the CD41/CD42 population separated from the progenitor cells.

A: Representative flow cytometry plots are shown in Supplementary Figure S1g. The CD41⁺/CD42⁺ populations can be found in the plots as green dots. Based on the plots from the ALI and control groups, the mature megakaryocyte population did not change significantly in shape but was obviously augmented. Per the reviewer's suggestion, we showed separate CD41⁺/CD42⁺ plots as well.

Q: 1E. Serum TPO levels. Unchanged in treated mice, however described in text as reduced. Unclear why levels increased in control mice?

A: We apologize for the unclear description. Reduced TPO means that the level of TPO in the ALI group is lower than that in the control group, which must be the consequence of liver injury. In regard to the absolute serum TPO level, it did increase in the control mice during the detection period. A possible reason for the TPO upregulation is the tail vein incision and the 100 μ L blood harvest that were performed every other day for all the tested mice; such volume and frequency bleeding might alter the TPO expression needed for thrombosis. Notably, the 100 μ L blood sample from each mouse was only needed for the ELISA detection. For the other set of experiments, only 20 μ L of blood was harvested from each tested mouse. Strict controls were used in all the experiments to add to the credibility of the studies.

We will change the description in the relevant results section in the new submission for improved clarity.

Q: I appreciate the experiments that Qu and colleagues carried out, however the authors did not assess RHOB and miR-1915-3p expression to confirm their role in modulating PMP function and megakaryocyte differentiation following the in vivo experiments. As the authors confirmed that decreasing miR-1915-3p expression attenuated CD41+/CD61+ populations in the cell lines (fig 4), key experiments would be to assess RHOB protein and gene expression in megakaryocytes derived from the irradiated mouse model untreated or treated with PMPs. RHOB activity should also be assessed in purified megakaryocytes from this model. miR-1915-3p expression should also be analysed in megakaryocytes in the mouse models following PMP injection.

Similarly, the above-mentioned experiments should be carried out in the ALI mouse model that demonstrated thrombocytosis linked to elevated PMP formation and megakaryocyte differentiation and maturation. The authors need to demonstrate that RHOB gene and protein expression is attenuated in this model and possibly RHOB activity.

A: We appreciate the reviewer's suggestions. As mentioned above, we discovered miR-1915-3p in mice through deep sequencing and bioinformatics analysis. Based on this, we have improved our mechanistic studies in vivo. Additional experiments, including qPCR and Western blotting, were performed to determine the RHOB and miR-1915-3p expression in vivo under ALI conditions. The response of RHOB and miR-1915-3p from the irradiated mice under PMP treatment was also assessed. By these studies, we confirmed the positive effect of PMPs on megakaryocyte differentiation and maturation in vivo with the two mouse models and confirmed the mechanism we discovered in vitro.

Q: FIG 2. Where is the ploidy data?

A: We apologize for the lack of a ploidy analysis. We treated the CD61-positive pro-MKs with PMPs and show polyploidy populations in Figure 2g and Supplemental Figure S3e.

Q: FIG3. The changes observed in D are small and the trend look the same for all miRs included and not specific to miR-1915-3 p.

A: The miRNAs we tested in Figure 3D were all chosen due to the enriched expression in PMPs and the capability to be transferred to hematopoietic stem/progenitor cells. Accordingly, it is possible that all these miRNAs have a positive effect on megakaryocyte differentiation, and their transfection resulted in a similar trend of megakaryocytic surface marker expression in the cell lines we tested. The transfection was repeated 3 times and statistically analyzed. Among these miRNAs, miR-1915-3p showed consistent and profound effects in all experiments. Therefore, we chose miR-1915-3p for further experiments. However, we could not exclude the positive effect of other miRNAs on megakaryocyte differentiation by this state.

Q: Row 72 “and release the mechanism underlying” What does this mean?

A: We apologize for the grammar issues. We indicated the elucidation of the underlying mechanism.

Q: Can PMPs be pre-treated with an RNase prior to injections? if it is impossible to target miR-1915-3p expression then this should be considered and should be included as a control when treating the irradiated or ALI mice to validate at least in part, that a micro RNA has a role in this mechanism.

A: We thank the reviewer for the suggestion. However, it is difficult to transfer RNase into PMPs without breaking the intact PMPs. We tried to downregulate miR-1915-3p by miR-1915-3p inhibitor transfection, but the effect was not satisfactory. The high CG content in miR-1915-3p may be responsible for these results. However, we agree

that it is important to confirm the role of miR-1915-3p in vivo. For this aim, we first confirmed in the ALI model and irradiated mouse model that the level of miR-1915-3p could be upregulated in bone marrow nucleated cells when the PMPs increased. Then, we modulated miR-1915-3p expression in the PMPs by transfecting miR-1915-3p mimics into PLTs and injected these miR-1915-3p-PMPs into irradiated mice to verify the important role that miR-1915-3p plays in thrombopoiesis of irradiated mice. An increased proportion of whole megakaryocytic cells and megakaryocyte-erythroid progenitors (MEPs) in the bone marrow and an increased number of PMPs in the peripheral blood were also detected in the mice injected with miR-1915-3p-PMPs on day 14, suggesting that overexpression of miR-1915-3p in PMPs promoted the expression of MK integrins and the production of platelets in vivo (Figure 6j-k, Supplemental Figure S10d).

Q: Overall there is an inconsistent size of graphs and font in figures and throughout the text. Additionally, grammatical errors are littered throughout the article that need to be addressed.

A: Thank you very much for noting the graphs and grammatical issues in our manuscript. According to your comments, we rearranged the figures for improved clarity and polished the manuscript with professional assistance.

Minor comments:

A: We thank the reviewer for his/her careful review. We have made changes according to the comments raised below.

Row 46 “thrombocytosis” should be replaced with thrombosis.

We have corrected this in the new submission.

Figure 1A: no units are labelled on the y-axis.

As mentioned above, the platelet counts are shown as relative levels. New units were labeled as %.

Figure 1B: Acute liver injury should be labelled.

These pictures present the states of the bone marrow after acute liver injury. We have already labeled “acute liver injury” in the figure. The injury to the liver is shown in Figure S1a and S1b.

Figure 1C: Percentage of CD41+ cells on Y axis.

The new Y axis was labeled as the percentage of BM nucleated cells (%).

Figure 1G: In the control vs ALI mouse model, the expression of PMPs was assessed over 6 days. The authors stained for CD41, and CD62 to determine the microparticles from activated platelets, however CD62+ needs to be labelled on fig 1G as only CD41+ MPs may refer to megakaryocyte MPs.

We apologize for the confusion in this section. The line graph in Figure 1 g shows the percentage of CD62p⁺CD41⁺ PMPs in CD41⁺ MPs. The new Y axis in Figure 1i (new submission) was labeled CD62p⁺CD41⁺ PMPs in CD41⁺ MPs (%).

Figure 1 legend; (a) During CCl4 induced acute liver injury (ALI).

We have made the correction accordingly.

FIG 1 legend. Progenitor cells are described as “neonatal” but they have used adult mice?

We apologize for the mistake; “neonatal” has been deleted.

Figure 2: Image quality/resolution is poor

We have improved the quality of the image in the new submission.

Figure 3 A: heat map needs to be revised, the labels are illegible, perhaps not all the miRNAs need to be shown.

The heat map has been revised. Only highly expressed miRNA contents of the PMPs from five different donors are shown.

Figure 4: legend title- Altering the expression of miR-1915-3P effects MK differentiation and maturation

We have the title changed accordingly.

Figure 5 C, F, G: y axis should state gene expression relative to Gapdh

The y-axis has been revised.

Supplementary data:

Figure S1B: haemoglobin units are not stated

Similar to the platelet counts, the hemoglobin units are also shown as relative levels. New units were labeled as %.

Figure S1C: The percentage for double positive CD42/CD71+ population should be included.

The percentage of the double-positive CD42⁺/CD71⁺ population is shown.

Figure S7 is missing.

We did have Figure S7 added. This issue might be because of the integration problem during the paper transmission. We will carefully review all the documents in the new submission.

Figure S8: A description of the schematic figure needs to be included describing the proposed mechanism of megakaryocyte differentiation independent of TPO.

We have added this information to the new submission.

Reviewers' comments:

Reviewer #1 (Remarks to the Author):

This a revised manuscript by Qu and colleagues. The authors made efforts to improve the manuscript and now provide sufficient information regarding PMP preparation, isolation and characterization. Their hypothesis is interesting and the manuscript is much improved.

There are however concerns that were not fully addressed:

1) The authors now provide sufficient information on methodologies used to isolate PMP and now carefully characterize them. In agreement with the current literature, however, I suggest that the author use the terminology "extracellular vesicles", rather than microparticles.
2) Although the authors justified their statistical analyses and now provide P values, most of the effects measured are very modest and it is often hard to believe they can be statistically significant (and they are unlikely to be biologically relevant)

a. Rather than supporting their statistical analyses, as it was asked in the previous reviews, the authors mention that Figure 1C was erroneously labeled, but seem to present the same data using a different graph representation.

b. It is also unclear how repeated-measures one-way ANOVAs were used as mice had to be sacrificed to quantify MK in bone marrow.

c. There remain concerns in Fig 4C (effect extremely modest with 200-fold overexpression by transfection) and with the platelet count in mice (Fig. 6F, presented as ratio), and elsewhere throughout the manuscript. In most experiments, I was not convinced by the rigor of the biostatistical analyses.

d. Platelet counts must be presented as number/volume

Reviewer #2 (Remarks to the Author):

THE AUTHORS HAVED ADEQUATELY ADDRESSED MY CONCERNS

Reviewer #3 (Remarks to the Author):

Qu and colleagues have responded with more thorough evidence for the role of miR-1915-3p in megakaryocyte differentiation and platelet production in a TPO independent manner. However, I have some additional comments.

Q: FIG 1. Fig 1A. I suggest including actual platelet counts. Not sure what units are used at the moment

A: Similar to the question raised before, the platelet counts are shown as relative levels. We have shown the raw data in Supplemental Figure S1d.

New response:

Thanks for including platelet counts as raw data in FIG S1d. However, only D8 is included.

Please include full data set with all timepoints indicated in FIG1A. The number stated for control mice (~500 x10⁶/ml) is not within normal range. This would be considered thrombocytopenia in a WT mouse. Please clarify this. Did the vehicle cause thrombocytopenia? Did you also include an untreated control group?

Minor comments:

- The ploidy data in fig 1d is not referred to in the text.
- Supp figure 1f- the label for CON and ALI next to fig 1f needs to be swapped.
- Scale bar inconsistent in graph and fig legend for figure 2a.
- Figure 2f -what is the y axis?
- In the text you refer to both figure 3a and b as Venn diagrams, add heat map also.
- Is fig 6l Rhob protein levels graph supposed to be saline vs PMP? It currently says CON vs ALI.

We would like to express our sincere appreciation to the reviewers for the constructive and positive comments.

Replies to reviewer 1

This a revised manuscript by Qu and colleagues. The authors made efforts to improve the manuscript and now provide sufficient information regarding PMP preparation, isolation and characterization. Their hypothesis is interesting and the manuscript is much improved.

There are however concerns that were not fully addressed:

1) The authors now provide sufficient information on methodologies used to isolate PMP and now carefully characterize them. In agreement with the current literature, however, I suggest that the author use the terminology “extracellular vesicles”, rather than microparticles

A: We thank for the reviewer’s suggestion. However, term “PMPs” is more highly literature supported, and we prefer to use it instead of “extracellular vesicles”. Here are some references we quoted for supporting:

Apart from the release of secretory vesicles by specialized cells, which carry, for example, hormones or neurotransmitters, all cells are capable of secreting various types of membrane vesicles, known as extracellular vesicles, comprising exosomes (30-100nm) and microparticles (50-1000nm, also known as microvesicles). Transmission electron microscopy images and flow cytometry plots showed that isolated PMPs were heterogeneous and spherical, with diameters ranging from 100 to 1000 nm. (Shedding light on the cell biology of extracellular vesicles. Nat Rev Mol Cell Biol, 19(4), 213-228.)

Exosomes are mostly produced in normal cell state. In contrast, we activated platelets with calcium and thrombin in this study. Upon activation, platelets are extremely potent at producing microvesicles, which are historically known as microparticles. (The diversity of platelet microparticles. Curr Opin Hematol (3.331), 22(5), 437-444.)

For these reasons, platelet-derived microvesicles are commonly referred to as “microparticles”, and this terminology will be used throughout this article.

2) Although the authors justified their statistical analyses and now provide P values, most

of the effects measured are very modest and it is often hard to believe they can be statistically significant (and they are unlikely to be biologically relevant)

A: We have consulted a lecturer of the Statistics Department of our academy and updated our statistical methods according to his suggestion.

a. Rather than supporting their statistical analyses, as it was asked in the previous reviews, the authors mention that Figure 1C was erroneously labeled, but seem to present the same data using a different graph representation.

A: In the first submission, we made mistake, put the plots of day 4 to the site for day 8 in figure 1C. The same plots were showed twice. We have corrected the mistake in previous revision. For the reviewer's convenience, the correct plots in that version are attached below. We'd like to point out that even the units in y-axis are different for these two time-point plots. They are really two set of data from day 4 and day 8. We have changed the data representation and taken more appropriate statistical method in the latest submission.

b. It is also unclear how repeated-measures one-way ANOVAs were used as mice had to be sacrificed to quantify MK in bone marrow.

A: The unclear description of the study method might result in the confusion of the reviewer. We are sorry for the blur. Correctly, two tailed unpaired t-tests were used as mice had to be sacrificed. The specific statistical analysis chosen for each experiment was illustrated and described in relevant figure legends.

c. There remain concerns in Fig 4C (effect extremely modest with 200-fold overexpression by transfection) and with the platelet count in mice (Fig. 6F, presented as ratio), and elsewhere throughout the manuscript. In most experiments, I was not convinced by the

rigor of the biostatistical analyses.

A: We thank for the reviewer's carefulness. But as we explained previously, different subjects used in these experiments resulted in different biostatistical analyses and data interpretation.

In Fig 4C, 24 hours after transfection of miR-1915-3p mimics, the 200-fold overexpression was confirmed by qPCR. The expression of the MK biomarkers (CD41 and CD61) elevated during megakaryocytic induction. However, the subjects used in this study were the primary cells obtained from unique cord blood donors. The individual difference resulted in high standard deviations and lowered the effect display. Considering the subjects in this study, it might be inappropriate to present the data with linear continuous Y-axis. To better and more correctly display the changes of megakaryocytes surface markers, we separately presented the percentage of CD61+ and CD41+/CD61+ on 15 day after transfection and showed the data as mean±S.E.M. instead of mean±S.D. (The same change was made in Figure 2c, Supplementary Figure3b, Figure 4c, Supplementary Figure 6f).

In Fig 6F, the count of platelet was presented as number/volume per reviewer's suggestion. The statistical methods of some experiments have also been modified.

d. Platelet counts must be presented as number/volume

A: We thank for the reviewer's suggestion. Count of platelet, white blood cells and red blood cells, neutrophils and monocytes are presented in the new submission. Two-way repeated measures ANOVA with Bonferroni's multiple comparison tests were used. (Figure 1a, Supplementary Figure1d, Figure 6f, Supplementary Figure 9h, Supplementary Figure 10d)

Replies to reviewer 2

THE AUTHORS HAVED ADEQUATELY ADDRESSED MY CONCERNS

A: We thank for the reviewer's appreciation and recommendation.

Replies to reviewer 3

Qu and colleagues have responded with more thorough evidence for the role of miR-1915-3p in megakaryocyte differentiation and platelet production in a TPO independent manner.

However, I have some additional comments.

Q: FIG 1. Fig 1A. I suggest including actual platelet counts. Not sure what units are used at the moment

A: Similar to the question raised before, the platelet counts are shown as relative levels.

We have shown the raw data in Supplemental Figure S1d.

A: We have made change according to the reviewer's recommendation. Count of platelet, white blood cells and red blood cells, neutrophils and monocytes are presented in the new submission. Two-way repeated measures ANOVA with Bonferroni's multiple comparison tests were used for statistical analysis. (Figure 1a, Supplementary Figure 1d, Figure 6f, Supplementary Figure 9h, Supplementary Figure 10d)

New response:

Thanks for including platelet counts as raw data in FIG S1d. However, only D8 is included.

Please include full data set with all timepoints indicated in FIG1A. The number stated for control mice (~500 x10⁶/ml) is not within normal range. This would be considered thrombocytopenia in a WT mouse. Please clarify this. Did the vehicle cause thrombocytopenia? Did you also include an untreated control group?

A: We appreciate the reviewer for the perspective suggestion. Non-treated mice (WT) were set as baseline control and the count of platelet showed no significant difference between WT and CON mice. Full data with all timepoints was presented in the new submission. In previous submission, we did have platelet baseline checked but just for one time point. The platelet counts were indeed around 500 x10⁶/ml in WT mice. It might be because of different animal batch that showed different platelet baseline. (However, the trend of platelet alteration under PMP treatment was consistent.) To make the data more convincing, we have repeated the experiment and showed a new set of data in this submission.

Minor comments:

A: We would like to firstly thank for the carefulness of the reviewer. Changes have been made

accordingly.

- The ploidy data in fig 1d is not referred to in the text.

A: We are sorry for missing the description of fig 1d, it has been added in the new submission.

- Supp figure 1f- the label for CON and ALI next to fig 1f needs to be swapped.

A: We have swapped the label for CON and ALI in Supplementary Figure 1f.

- Scale bar inconsistent in graph and fig legend for figure 2a.

A: We have corrected the wrong scale bar in figure legend. It is 5 μm .

- Figure 2f -what is the y axis?

A: We have the missing y axis added. It is Cell Diameter (μm).

- In the text you refer to both figure 3a and b as Venn diagrams, add heat map also.

A: We have added “heat map” in the text.

- Is fig 6I Rhob protein levels graph supposed to be saline vs PMP? It currently says CON

vs ALI.

A: The unclear description in text might result in the confusion of the reviewer. We have changed the description from “Compared to the control treatment” to “Compared to the Saline treatment”.

REVIEWERS' COMMENTS:

Reviewer #1 (Remarks to the Author):

The main concerns are addressed

Reviewer #3 (Remarks to the Author):

The authors have addressed my concerns.